# Barium content of Archaean continental crust reveals the onset of subduction was not global

Guangyu Huang [1] ✉, Ross N. Mitchell [1,2] ✉, Richard M. Palin [3], Christopher J. Spencer [4] & Jinghui Guo[1,2]

Earth's earliest continental crust is dominated by tonalite–trondhjemite–granodiorite (TTG) suites, making these rocks key to unlocking the global geodynamic regime operating during the Archaean (4.0–2.5 billion years ago [Ga]). The tectonic setting of TTG magmatism is controversial, with hypotheses arguing both for and against subduction. Here we conduct petrological modeling over a range of pressure–temperature conditions relevant to the Archaean geothermal gradient. Using an average enriched Archaean basaltic source composition, we predict Ba concentrations in TTG suites, which is difficult to increase after magma generated in the source. The results indicate only low geothermal gradients corresponding to hot subduction zones produce Ba-rich TTG, thus Ba represents a proxy for the onset of subduction. We then identify statistically significant increases in the Ba contents of TTG suites worldwide as recording the diachronous onset of subduction from regional at 4 Ga to globally complete sometime after 2.7 Ga.

Tonalite–trondhjemite–granodiorite (TTG) suites and their metamorphosed equivalents (TTG gneisses) are the dominant components of Archaean continental crust[1,2]. TTG suites are typically interpreted as the product of partial melting of a hydrated basaltic source[3]. Most TTG suites in Archaean cratons occur as deformed gneisses and have consistent and simple mineralogy, but their diverse compositional characteristics (e.g., Sr/Y, La/Yb, and Nb/Ta) have led to a variety of models for how Earth's earliest continental crust formed and subsequently evolved through time. Among these continental-crust-forming magmatic models, partial melting of subducted slabs or the lower portions of thickened mafic crust are the most widely cited mechanisms for the generation of TTG magma[4,5]. Exploring the rate of change of crustal growth, the petrogenesis of TTG magmas, and secular changes in TTG compositions may provide insights into the geodynamic settings of their formation and thus, potentially constrain the age of the onset of global subduction[2–12].

TTG suites are typically rich in silica (commonly $SiO_2 > 70$ wt.%), have $K_2O/Na_2O < 0.5$, and have low concentrations of ferromagnesian elements ($[MgO + FeO + TiO_2 + MnO] < 5.0$ wt.%), with an average Mg# (= atomic Mg/[Mg + Fe] × 100) of 43 (ref. 2; Fig. 1a, b). However, neither experimental petrology nor thermodynamic modeling has been able to satisfactorily simulate the production of such high-MgO TTG suites solely by the partial melting of a basaltic protolith[13–15]. Some MgO-rich melts or rocks (e.g., komatiite, picrite, and peridotite) may need to be assimilated into a TTG magma or into the source rock to produce the observed compositions[16,17]. Although trace element contents vary between magmatic systems, in general, TTG suites are rich in light rare earth elements (LREE), but depleted in heavy rare earth elements (HREE) and/or strongly compatible elements, such as Cr and Ni, which have average contents of 40 and 18 ppm, respectively[2]. Experimental petrology has shown that partial melts of amphibolite and eclogite are identical to TTG suites in terms

[1]State Key Laboratory of Lithospheric Evolution, Institute of Geology and Geophysics, Chinese Academy of Sciences, 100029 Beijing, China. [2]College of Earth and Planetary Sciences, University of Chinese Academy of Sciences, 100049 Beijing, China. [3]Department of Earth Sciences, University of Oxford, Oxford OX1 3AN, UK. [4]Department of Geological Sciences and Geological Engineering, Queen's University, Kingston, ON K7L 3N6, Canada. ✉e-mail: huangguangyu@mail.iggcas.ac.cn; ross.mitchell@mail.iggcas.ac.cn

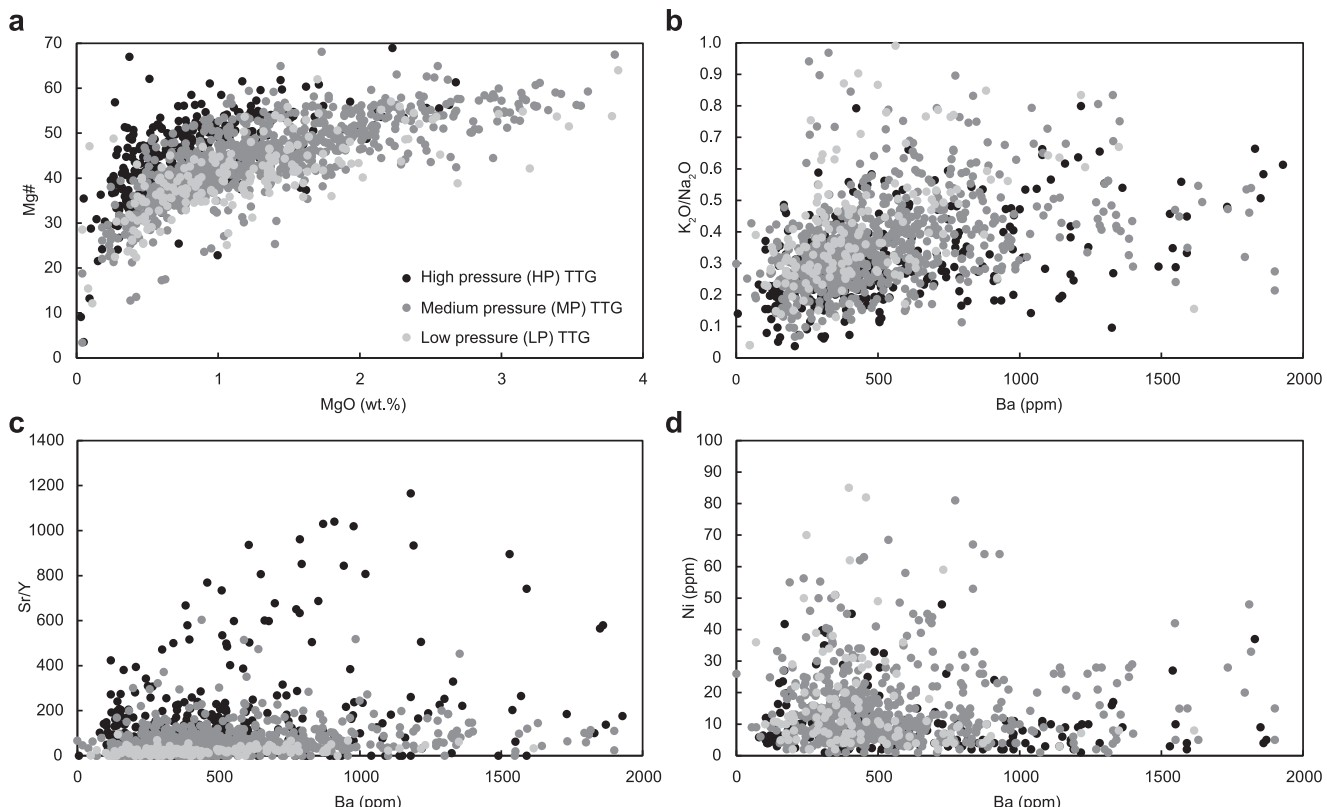

**Fig. 1 | Compositions of tonalite–trondhjemite–granodiorite (TTG) suites. a** Mg# versus MgO. **b–d** K₂O/Na₂O, Sr/Y, and Ni contents versus Ba contents. Data are from ref. 82.

of major oxides, such that TTG suites can be considered as the products of fluid-absent melting of enriched hydrated and metamorphosed basalt[13,18–23]. As important peritectic minerals in the residue that control melt trace element composition (e.g., HREE, Sr, and Y), garnet (rich in HREE) is stable at a higher pressure (>1.5 GPa) than plagioclase (rich in Sr), with this variation in residual mineralogy having been used to divide TTG suites into high-, medium- and low-pressure (HP, MP, LP) types based on their La/Yb and Sr/Y ratios[2,3]. HP TTG suites have high La/Yb and Sr/Y ratios, whereas LP TTG suites have low La/Yb and Sr/Y ratios, and MP TTG suites exhibit transitional characteristics[2] (Fig. 1c).

Nonetheless, the depth of crustal melting is not the only factor controlling the Sr/Y and La/Yb compositions of the melt. Recent work suggests that continental-crust-forming magmatism during the Archaean was likely more complex than simple, single-stage fluid-absent melting[10,11,24–35]. Fluids would have been abundant in hot subduction zone settings[10,24–26], and greenstone belt material may have "dripped" into the underlying mantle to introduce considerable amounts of fluids, where such vertical motion in the Archaean Eon may differ from the dominantly horizontal motion of plate tectonics[11]. Plagioclase was likely efficiently consumed during fluid-present melting at low pressure and temperature conditions, leading to the apparent generation of "HP TTG suites"[27]. Further, crustal melting may also be accompanied by melt extraction, ascent, and emplacement, during which crustal assimilation and/or fractional crystallization and accumulation of peritectic minerals would inevitably change the original composition of the melts[28–32]. Appreciable amounts of plagioclase accumulation and fractional crystallization of hornblende would induce HREE depletion and raise Sr/Y in the melt, similar to that of HP TTG magma[32–36]. In the Barberton Greenstone Belt of South Africa and the Wawa Gneiss Domain of Canada, a diversity of Sr/Y ratios has been attributed to plagioclase accumulation in mushy mid-crustal TTG suites[34,35].

Given such considerable variation, the Sr/Y ratio of TTG suites is increasingly being considered as an unreliable indicator of the depth of crustal melting. Instead, a pressure-dependent indicator of the original melts that is not influenced by subsequent magmatic processes (i.e., assimilation of mantle rock and/or sediments, fractional crystallization, and accumulation) would be suitable for interpreting the tectonic setting of TTG magma formation. As an incompatible element, Ba has a rather low concentration in MgO-rich rocks that are often quite primitive. Based on mass balance calculations, the assimilation of MgO-rich rocks into a magmatic system would not cause an increase in Ba content of the mixed magma. However, the assimilation of sediments, rich in large ion lithophile elements (LILE; i.e., K, Rb, Sr, Ba, etc.), could raise the Ba content in TTG magma[37]. If the assimilation of sediments were to explain elevated Ba enrichment in TTG magma, then their LILE and Ba contents should be coupled. However, most high-Ba TTG suites are characterized by low K₂O/Na₂O (Fig. 1b), as well as low Rb. This compositional decoupling thus excludes the possibility of sediment assimilation as a general mechanism accounting for Ba enrichment. Even though plagioclase accumulation would efficiently change the Sr/Y ratio in TTG magma, it cannot raise the Ba content of a melt, as the partition coefficient between plagioclase and granitic melt is very close to 1 (refs. 4, 35; Supplementary Table 1). As an illustration, the Palaeoarchaean Barberton trondhjemite suites are suggested to have differentiated from tonalitic magma generated at <40 km depth, where plagioclase accumulation leads to a diversity of TTG, with distinct K₂O/Na₂O, Sr/Y, and La/Yb but constant Ba contents[34]. Of all rock-forming minerals likely involved in Archaean crust formation processes, only biotite exhibits strong compatibility with Ba; however, biotite accumulation during TTG melt production, ascent, and crystallization is also unlikely until the very last stages of magmatic evolution[34]. Importantly, Ni is also compatible in biotite, but Ni contents in most high-Ba TTG suites (Ba > 1000 ppm) does not exceed 30 ppm, which is even lower than those of low-Ba TTG suites (Fig. 1d).

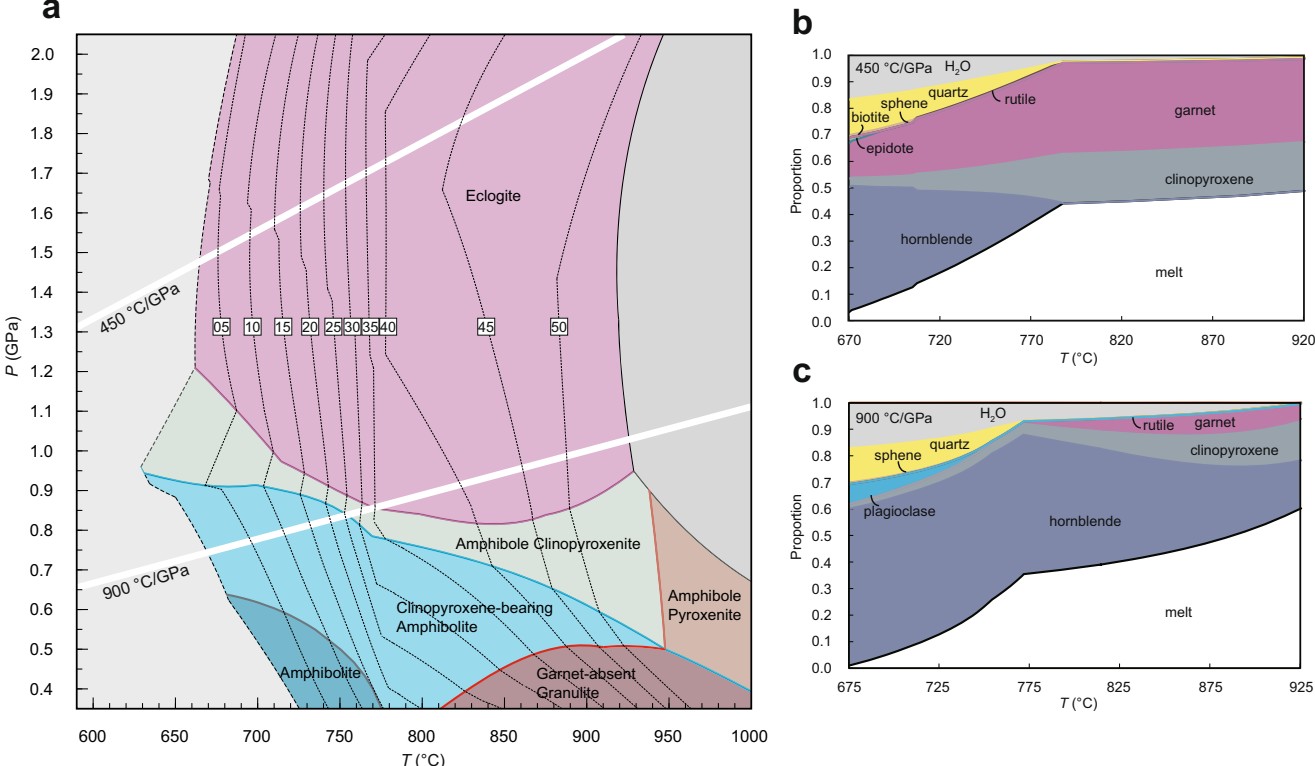

**Fig. 2 | Phase diagrams for an average Archaean tholeiitic basalt. a** A simplified pressure–temperature (*P*–*T*) diagram of an average Archaean tholeiitic basalt composition, labeled with the mol.% of melts. **b**, **c** mineral and melts proportions versus temperature, along 450 °C GPa⁻¹ and 900 °C GPa⁻¹, respectively.

Therefore, crystal accumulation cannot explain these compositional features, and the only remaining possibility would be a diversity of original melt compositions.

In this work, we present and test a compositional proxy−barium (Ba) contents in TTG suites−for interpreting the tectonic setting of TTG magma formation by conducting petrological modeling of partial melting of a basaltic composition[26]. It has been noted that TTG suites are too enriched in LILEs to be generated from mid-ocean ridge basalts (MORB) and oceanic plateau basalts (OPB), which are depleted or not enriched enough in LILEs[26,38]. Enriched oceanic island basalts (OIB), even though they are enriched in LILEs[26], were likely too rare in the Archaean to be a viable source for large volumes of TTG magma. Ge et al.[26] compiled more than 1,000 analyses of enriched Archaean tholeiitic basalts that are characterized by moderately enriched LILE and have Th/Nb >0.1 and suggested these basalts would be a suitable source for generating large volumes of TTG magmas. These basalts have a wide range of trace element compositions, and would not necessarily be related to subduction since the oblique array in the Th/Yb-Nb/Yb plot joining a rather depleted portion of the mantle array to the arc field would result from continental-crust assimilation[39]. To verify the hypothesis that the diversity of original melt compositions accounts for high-Ba TTG magmas, the average enriched Archaean tholeiitic basalt composition[26] is used to carry out petrological modeling at different geothermal gradients and water contents, which allowed predictive modeling of melt compositions in different geodynamic settings. The average documented Ba content of enriched Archaean tholeiitic basalts is 107 ppm, with a median value of 62 ppm. Among the more than 1000 Ba content data, nearly 90% of these basalts have less than 160 ppm, indicating that there is not enough variability in the primary Ba contents of Archaean basalts to account for the diversity of Ba contents in TTG suites, especially in the early Archaean when general compositional diversity (e.g., in Ba) is predicted to have been lower[26].

## Results and discussion

### Petrological modeling of Ba contents in TTG melts

Given that hot subduction and lower crustal drips are the two possible end-member geodynamic settings for Archaean TTG magma generation, two typical geothermal gradients of 450 °C GPa⁻¹ and 900 °C GPa⁻¹, respectively, are considered for investigating the evolution of the Ba content of the melts during metamorphism[2,40–42]. As an additional possibility, a geothermal gradient of 250 °C GPa⁻¹, corresponding to a cold subduction setting[40,43], is also explored. According to the assumption that fluid-absent melting is the main mechanism to generate TTG-like melts[15], previous work has explored the evolution of melt composition under water-unsaturated conditions. To investigate the evolution of Ba content in melts under water-saturated conditions, based on the possibility of fluid-present melting described earlier and to make a comparison with melts under water-unsaturated conditions, a pressure–temperature (*P*–*T*) diagram for an enriched Archaean tholeiitic basalt composition is produced with an H₂O content of 7.0 wt.% to ensure all the calculated melts are generated through fluid-present melting (Fig. 2a and Supplementary Table 2). Compared with the *P*–*T* diagrams calculated at water-reduced conditions[15] (Supplementary Figs. 1–4), the rocks with higher water contents are much more fertile (Fig. 2b, c). For the hot subduction setting, approximately 50 vol.% of melt is predicted to be generated per unit volume of protolith if sufficient water is present, compared to ~20 vol.% in a water-limited environment. High water contents also shrink the stability field of hornblende and quartz to lower temperatures, and these minerals are preferentially and efficiently consumed at relatively low temperatures (Fig. 2b). For an Archaean drip setting, plagioclase stability would be shrank to lower pressure and quartz to lower temperature (Fig. 2c). While amphibole tends to remain stable before quartz and plagioclase are consumed. These mineralogical differences in residue lead to a diverse range of melt composition evolutions. For a cold subduction setting,

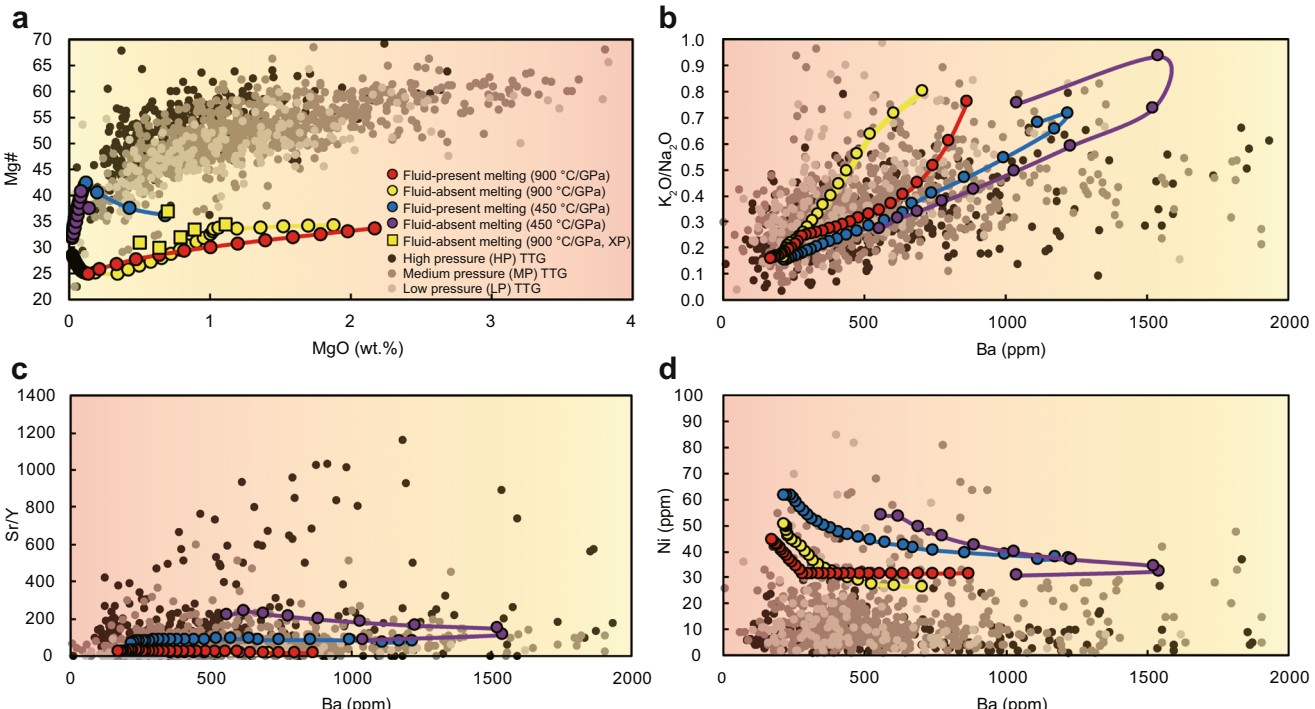

**Fig. 3 | Modeled tonalite–trondhjemite–granodiorite (TTG) melts compositions. a** Mg# versus MgO content of modeled melts, experiment data from similar protolith composition was also plot for comparison[13]. Numbers in the legend show the thermal gradient. XP indicates experimental results. **b–d** K$_2$O/Na$_2$O, Sr/Y, and Ni contents versus Ba contents, concordant with natural samples in Fig. 1. The circles represent the calculated melt fractions at 2 mol.% intervals. The background color change from yellow to orange (cold to hot, respectively) indicates the direction of temperature increase.

calculated melt proportion isopleths lie almost parallel to the geothermal gradient, leading to no more than 4 vol.% of melt generation (Supplementary Fig. 5), which could not be extracted from the protolith[44]. Such results thus rule out cold subduction for TTG magma generation.

Predicted melt compositions are clearly a function of the *P–T* conditions of generation, as well as their protolith compositions, including water content. Given a similar protolith composition, the modeled melts generated through fluid-absent melting at higher thermal gradients (i.e., Archaean drip environment) are identical to those produced in experiments under similar conditions[13] (Fig. 3a). Fluid-present melting experiments are relatively rare, especially at high pressure, given kinetic limitations to achieving equilibrium. Even so, recent experiments[14] on a low-Mg basaltic protolith (Mg# =38) also show similar results with petrological modeling[15] (Mg# =36) (Supplementary Data 1). For specific Archaean basaltic compositions, the melt compositions produced at different thermal gradients are different when partial melting begins, even though they converge at higher degrees of anatexis. At all conditions, the early melts are usually potassium-rich (granitic) and tend to reach a TTG-like composition above ~7 mol.% melt. Low degrees of partial melting (~7–9 mol.% of melts) at lower thermal gradients (i.e., hot subduction environment) would produce more Ba-rich melts (>1000 ppm), regardless of water content (Fig. 3b, c, d and Supplementary Data 2). Fluid-absent melting is expected to generate the highest Sr/Y melts, being two times higher than fluid-present melting and ten times higher than those produced at higher thermal gradients (Fig. 3c). All melts are calculated to have low Ni contents (<70 ppm), which are negatively correlated with Ba content (Fig. 3d), consistent with natural TTG suites.

## Ba in TTG suites as a proxy for early Earth tectonic setting

It is widely accepted that the mantle potential temperature in the early Archaean was higher (~250 °C) than that at present[45,46]. Secular

cooling of the ambient mantle would influence coupling across the lithosphere-asthenosphere boundary, possibly leading to a transition in the tectonic regime from a stagnant lid to modern plate tectonics[47]. Even so, the time period corresponding to this transition is debated, with propositions ranging from the Eoarchaean to Neoproterozoic[26,48–50]. Stagnant-lid regimes differ from plate tectonic (mobile lid) regimes in being dominated by vertical tectonic motion and deformation (e.g., dome and keel; sagduction; drip), as opposed to horizontal tectonic features (e.g., linear collisional orogens). Well-preserved vertical structures can be recognized at some older cratons, such as eastern Pilbara craton in western Australia, Kaapvaal and Zimbabwe cratons in southern Africa[51–53], although individual cratons do show a succession of "dominantly vertical" followed by "dominantly horizontal" regimes throughout their evolution. Geodynamic modeling shows that the *P–T* paths of rocks within these vertical structures follow a relatively high geothermal gradient[42]. Based on our calculated results, it seems plausible that only subduction zones would provide a sufficiently low geothermal gradient to produce Ba-rich magmas. Ba contents in TTG suites would not be influenced during fractional crystallization and magma assimilation (Supplementary Fig. 6), and the diversity of Ba contents in the source (enriched Archaean tholeiitic basalt) would be limited as described. Thus, it would be a sensitive indicator of the tectonic setting of formation.

Whenever a compositional proxy for subduction is introduced, its broader implications must be considered within a multi-proxy context. Based on different proxies, however, the timing of the onset of plate tectonics is contentious[48–50,54–56]. Most of the proxies either record the first appearance of specific rock types or metamorphic facies (e.g., blueschist, Precambrian paired metamorphism)[50,57,58], or indirectly infer the nature of continental crust[48,49]. Such data are usually considered as the upper limit age of onset plate tectonics. Compositional changes in Archaean basalts and komatiites might suggest a

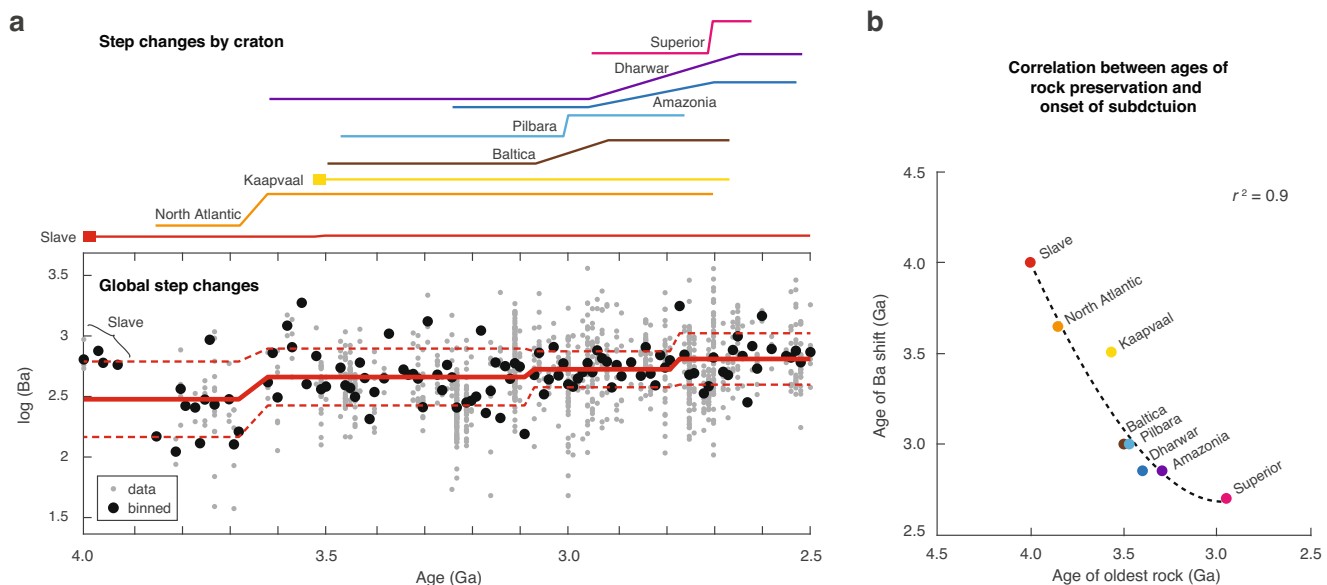

**Fig. 4 | Ba content of Archaean tonalite–trondhjemite–granodiorite (TTG) suites. a** Ba contents of TTG suites through the Archaean. The gray and black solid circles represent the real and binned compositions of natural TTG suites, respectively. Data are from ref. 6. **b** Correlation between age of Ba step change with the oldest rock by craton (degree 2 polynomial regression).

re-enrichment of Earth's mantle at 3.2 Ga[54], but dating such silica-undersaturated rocks is not as easy as felsic rocks. In contrast, the oldest granitoids on Earth can be precisely dated and formed at ca. 4.0 Ga in Slave craton (Acasta TTG suites[59]). Eoarchaean to Palaeoarchaean (4–3.2 Ga) TTG suites also occur in many cratons worldwide (e.g., Greenland, Barberton, and Pilbara), providing more information on the earliest Earth. Our data indicate that the Ba content in TTG suites is more closely controlled by initial partial melting than by later magmatic effects. We suggest such a compositional proxy (Ba) can provide the framework for developing a multi-proxy approach in the future.

## Secular change of Ba contents in Archaean TTG suites and implications for the onset of global subduction

Having suggested Ba contents as a proxy for the tectonic setting of TTG magmatism, we test for the presence of any potential shifts in the global record of Archaean TTG suites[6] that might indicate the evolution of geodynamic regimes (see Methods for details of the statistical change-point test used). Ba contents in TTG suites are unimodal and follow a lognormal distribution (Supplementary Fig. 7). As the Ba data span multiple orders of magnitude, log(Ba) is considered to assess the data as a time series (Fig. 4a). Recognizing the lognormal distribution is critical for establishing that high-Ba TTG suites are not outliers, but merely Ba-enriched end members.

Before 3.7 Ga, with the sole exception of Slave craton, nearly all TTG suites preserved in the rock record have low Ba contents, which indicates their formation along a hot geotherm. The Ba contents in TTG suites shows a sequence of three statistically significant positive shifts at 3.7, 3.1, and 2.8 Ga, implying a protracted and irreversible change took place in global geodynamics (Fig. 4a). Upon closer inspection, the ages of the positive shift in Ba when plotted for specific cratons are distinct (Fig. 4 and Supplementary Fig. 8). Thus, the presence of multiple "global" shifts actually reflects the fact that the Ba shifts from craton to craton are highly diachronous (Fig. 4a, b). The TTG suites of the Slave and Kaapvaal cratons preserve high-Ba values as early as 4.0 and 3.5 Ga, respectively, which they maintain throughout the rest of the Archaean. A strongly positive Ba anomaly has been identified at 3.5 Ga in Slave (Fig. 4a), which, if interpreted as an

indicator of subduction, is independently corroborated by a seismically imaged slab beneath the Slave craton dated to ca. 3.5 Ga (ref. 60). The ages of the various positive shifts within different cratons might result from different starting times of subduction. Subduction might have occurred in the Slave craton since 4.0 Ga, followed by initiation at 3.7 Ga in the North Atlantic craton and 3.5 Ga in the Kaapvaal craton, while most others initiated at 3.2–3.0 Ga (e.g., Baltica, Amazonia, and Pilbara), as late as 2.7 Ga (Superior craton), or even later for younger cratons like North China not analyzed herein. Ca. 2 Ga provides a lower bound for global subduction[61] whereas our study provides and upper bound sometime after 2.7 Ga.

The initiation of subduction would be a function of several factors, including mantle temperature, radiogenic heat production, density contrast between the lithospheric mantle and the convective mantle, and lithospheric thickness[36,62]. The whole Earth would not be homogenous in all these factors, so we may not expect the onset of subduction to be a globally isochronous event as has often been portrayed in past work. Subduction likely is initiated once all these factors were satisfied, which should naturally only be regional in scale at the beginning, and eventually evolve to a global scale once all regions eventually became favorable to subduction initiation[61–64]. Based on the notable change in the Ba content of TTG suites identified here, we instead propose a subduction propagation model, in which a transition from a stagnant-lid regime to plate tectonics would have occurred from regional (e.g., Slave and North Atlantic cratons) to global in scale. Such an age for the onset of plate tectonics is consistent with Earth's thermal history according to modeling and petrological data (Fig. 5). Because of uncertainties in observational inferences as well as theoretical models, the cooling history of Earth's mantle cannot tightly constrain when plate tectonics might have emerged from stagnant-lid convection (a putative pre-plate-tectonics regime) (Fig. 5), thus rendering the Ba proxy for subduction a critical constraint. Furthermore, the ages of Ba positive shifts of various cratons correlate strongly with the ages of the oldest rocks of each craton (Fig. 4b). According to our proposal that Ba positive shifts indicate subduction initiation, this correlation would thus imply an important relationship between rock preservation and subsequent subduction.

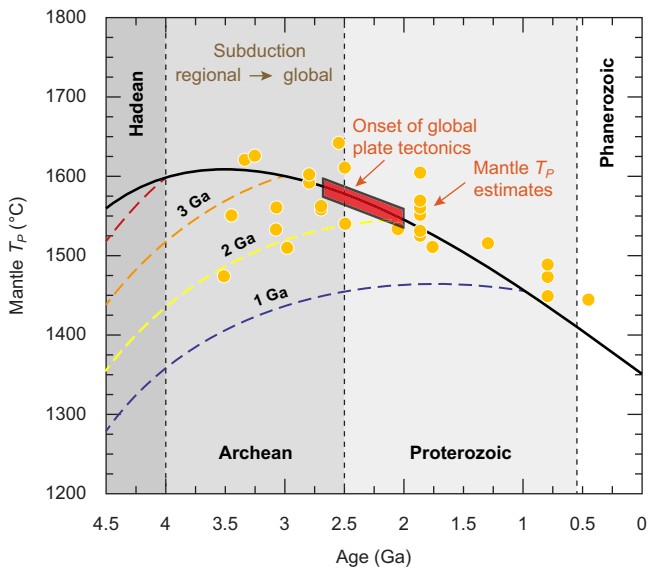

**Fig. 5 | Modeled thermal histories.** Modeled thermal histories with a switch in heat-flow scaling from stagnant-lid convection to plate tectonics at 1 Ga (blue), 2 Ga (yellow), 3 Ga (orange), and 4 Ga (red), while a case entirely with plate tectonics scaling is shown in solid (see ref. 83 for modeling details). Red box shows onset range for global subduction with 2.7 Ga as an upper bound (this study) and 2.0 Ga as a lower bound[61], consistent with estimates of mantle potential temperature ($T_p$)[45] as compared to thermal modeling.

## Methods

### Phase equilibria modeling

Phase diagrams were calculated using THERMOCALC v. 3.45 (ref. 65) with internally consistent dataset (ds62) of Holland and Powell[66]. The calculations were performed in $Na_2O$-$CaO$-$K_2O$-$FeO$-$MgO$-$Al_2O_3$-$SiO_2$-$H_2O$-$TiO_2$-$Fe_2O_3$ chemical system. The corresponding *a-x* models are clinoamphibole, augite and metabasite melt[67], orthopyroxene, garnet and biotite[68], plagioclase and K-feldspar[69], ilmenite[70], and magnetite[71]. Quartz and rutile are considered as pure phases.

### Accessory mineral solubility modeling

We assume that bulk rock compositions of Zr and P reside only in the zircon and apatite and do not substitute into other rock-forming minerals. Then the zircon and apatite retained in the residual would be calculated by solubility equations, along with initial bulk composition, stoichiometric concentrations of Zr and P in zircon and apatite and the degree of melting[72–75]. We use zircon solubility expression from ref. 76 and apatite solubility expression from ref. 77. Stoichiometric values of Zr in zircon and P in apatite are assigned as 497,664 ppm and 41 wt.%, respectively[72,78]. Degrees of melting are calculated by THERMOCALC.

### Trace element modeling

Trace element modeling was carried out using the batch melting equation $C_{melt}/C_{source} = 1/[D + F \times (1-D)]$[79], where $C_{source}$ and $C_{melt}$ represent concentrations of a trace element in the source rock and the resultant melt, respectively; $D$ ( $= \sum Kd \times X$) is bulk partition coefficient, where $Kd$ is mineral/melt partition coefficient and $X$ is the calculated mol.% of the mineral; $F$ is the degree of melting. The mineral/melt partition coefficients ($Kd$) used in the modeling are taken from literature[4].

### Statistics

To evaluate the possibility of state shifts, or change-point behavior, in the secular variability of the Ba data, we use a Bayesian change-point algorithm (conjugate partitioned recursion)[80,81] (Fig. 4a). The conjugate partitioned recursion algorithm uses a strategy of binary

partitioning by marginal likelihood combined with conjugate priors. The algorithm first calculates the marginal likelihood for both a no-change model and a change-point model to identify whether a change point is appropriate. If the marginal likelihood favors a change-point model, then the algorithm defines the change point and two-sigma uncertainty bounds of the two averages before and after the change point[81]. The analysis was conducted both on the global dataset and by craton. Then, the change points identified for each craton were also compared by degree 2 polynomial regression with the oldest rocks of each craton (Fig. 4b).

## Data availability

The mineral/melt partition coefficients ($Kd$) used in the modeling are tabulated in Supplementary Table 1. The bulk compositions used for phase equilibria modeling in this study are provided in Supplementary Table 2. The modeling results of the major and trace element compositions of the melts are provided in Supplementary Data 1 and Data 2, respectively.

## Code availability

The software and datafiles used to generate the phase diagrams presented herein can be downloaded at https://hpxeosandthermocalc.org/.

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

## Acknowledgements

We thank J. Korenaga for his help with the modeling of Earth's thermal history. This article was based on work supported by the National Natural Science Foundation of China (41890832, 42172217, and 41902057).

## Author contributions

G.H. and R.N.M. designed the project. G.H., R.N.M., and C.J.S. performed the modeling. G.H. wrote the original draft. R.M.P., R.N.M., C.J.S., and J.G. edited and commented on the draft.

## Competing interests

The authors declare no competing interests.
