## [Peer Review File · Nature Communications]

Barium content of Archaean continental crust reveals the onset of subduction was not globalREVIEWER COMMENTS

Reviewer #1 (Remarks to the Author):

Review of Nat. Comms manuscript “Barium content of Archaean continental crust...” (by Guangyu Huang et al)

The thought of using Ba in this way is quite an interesting a novel approach to a significant problem in how we view the (various and changing) origins of TTG and hence, earlier crustal evolution. There is a lot to like about this study and I certainly think it is a topic suitable for Nat. comms. The manuscript is generally quite well written but there are a few areas where arguments become a bit muddled – and where all of the listed co-authors perhaps could add an additional polish. I’m not convinced that the interpretation presented here is the only way to interpret that data presented, but it is new, certainly possible and will create interest to all struggling with ideas about how and when early crust forming processes operated. I would certainly like to see the idea put out there for wider consideration, after a bit of clarity on a few points and recommend publication with moderate revision.

One point that I have difficulty with is suggesting that a decrease in the LILE content of TTG at a point where subduction is suggested to become their new mode of occurrence is somewhat counter intuitive. You can’t attribute this all to a flat-subduction regime because you then deny the evidence for metasomatic enrichment in the mantle source of some Meso- to Neoproterozoic mafic sequences.

Also, you are putting a lot of trust in an element that is notoriously mobile, in TTG forming discernible correlation with basically nothing ...except, in some cases, for Sr! The fact that we see change points, and of a systematic direction itself is surprising and possibly quite meaningful.

Perhaps related to inherent Ba scatter - hopefully another reviewer is better able than I am to judge the validity of the statistics, but many of the change points shown in the sup data seem to be very tenuous. There does nevertheless appear to be a trend consistent with the given interpretation.

L22-23 – this is a bit of a nonsense – how can we have regional global subduction?

L32 – not really much ‘grey gneiss’ in any of the Australian Archean regions. L60-

62 – I certainly agree that this seems to true.

L72-73 - I seriously doubt that this is the case. I can accept that some Archean high Mg andesitic hybrids might form through interaction of small amounts of felsic material with much larger volumes of mafic or

ultramafic magma (e.g. Barnes and Van Kranendonk) but physical mixes of mafic magma into melts as felsic as the >70wt% silica melts you are talking about typically manifests themselves in the field very, very, clearly for what they are. Removing these from your dataset basically ensures that your argument about a lack of evidence for fractionation becomes a self-fulfilling prophecy. You might like to consider an alternative – that these higher Mg# samples have a more sanukitoid-like petrogenesis.

L 77-79 - this might be the case, but the further we go back in the Archean, the more 'sodic' systems become. If subduction was occurring in the early Archean, would we necessarily expect the same LILE (and particularly, K, Rb) enrichments then as we see today? Certainly more evidence exists in the Neoproterozoic, but this is when Ba concentrations in TTG apparently decrease!!

L87-88 – this apparent negative correlation between Ni and Ba is certainly not at all clear to me from this figure.

L96-98 – after quite a reasonably review of currently accepted views on TTG formation, it seems a bit surprising that water-saturated melting is now the sole consideration? This probably needs a line or two of justification!

L114 - I accept that we have to make such assumptions, but there are several Moyen papers that clearly articulate exactly how important source composition is.

L137 – . 160 ppm seem very low for TTG. The average Ba in 229 Pilbara TTG meeting the definition given above is 560ppm and for 1011 samples from the Yilgarn meeting these criteria Ba = 650ppm. I certainly agree that there is not enough variability in the primary Ba content of Archean basalt to account for the Ba contents of TTG and suggest that in many cases the source is clearly not that simple. I think this is a caveat you need to stress.

L139-140 – if this is indeed the case then the findings might be more relevant to the earlier Archean, when general compositional diversity (e.g. in Ba) is predicted to have been lower (reduces the issues of source variations).

L154 – I'm not sure that all of Mike Brown's data was restricted to the abovementioned regions.

L155-156 – not a sentence – and why would we expect a transition in Sr/Y through time related to plagioclase accumulation anyway?

L169-170 – using 'globally' here is very misleading given n=1 or 2. You have no clue that its global.

L174 - 175 – if the change is relevant only to the North Atlantic craton then don't refer to it as a global trend in the first place. This makes all of this paragraph very confusing. There is a very important and valid point to be made here, but it is being muddled.

L180-183 – so you are clearly NOT talking here about global subduction – this point needs to be made.

L183-187 – this is a misleading misinterpretation of what is said in REF11 – which in fact claims that at least some HP TTG might result through fractional crystallisation of primitive sanukitoid magmas – and proceeds to demonstrate this using a large dataset of temporally and spatially related samples that define a complete liquid line of descent!

203-204 – as it still is with the Ba proxy! i.e somewhere between 4 and 2.7Ga...or after!

L217 WHY do we need to search for a specific date!!!

This (the claimed pervasive nature of the required process – L185) does, however, highlight a particular point of relevance – that indeed perhaps supports the FC model for HP TTG rather than the one presented here – and that is that HP TTG (even using Ba concentrations as a proxy) are in fact not common, and in some cratons almost absent. For something that's heralding such a significant change in Earth dynamics, perhaps we would expect to see more of it preserved.

190-191 "Such an age for the onset...." What age? – you have given an age range, so where within that range are you putting the 'onset' – there is no real 'onset' and in all of this there is no indication of anything that clearly resembles a Plate tectonic regime of globally connected destructive margins supported by ridges.

Reviewer #2 (Remarks to the Author):

This paper tackles the topical subject of the conditions under which Archaean TTG were generated using petrogenetic modeling to estimate the Ba contents of melts generated under different thermal conditions. The minerals stable in the preferred basalt composition, and involved in partial melting, are sensitive to the geothermal gradients, and it is concluded that suitable Ba contents are generated at relatively low geothermal gradients. In some ways this is a confusing paper in that Ba contents are highlighted, and yet it is the thermal gradients that are key to the conclusion. That point has been made before by Tim Johnson and colleagues in Nature, and then, understandably, the key conclusions are expressed through trace element abundances. While Ba appears to work well, it is not clear whether there is something distinctive about Ba in this context, or other incompatible elements would have worked equally well. Th is often used as it may be less mobile during alteration and metamorphism.

Johnson et al used a broadly similar approach to argue that the first continents were not formed by subduction. It is unclear whether you are in agreement given the low Ba contents of the older TTG, or in disagreement in ways that are not clear to the reader, but it would be a great help if that was addressed.

It is striking that only two major geological settings were selected. The conclusion is that one works well, but that is not the same as saying it is the preferred solution given all the settings that might have been present in the Archaean – which is how the paper might be interpreted. Can it be shown that cold subduction can also be ruled out with this approach?

I am not sure of the source of the thermal conditions for hot subduction and drip tectonics, as references 33 and 34 are relatively geological. Yet the thermal conditions are key to the arguments presented, and how robust are the geothermal gradient estimates for the two settings?

I am sorry if I missed it, but it was difficult to find the minor and trace element pattern of the source basalt, and that would certainly help the geochemists. I suspect you might argue that it doesn't matter whether it is a subduction-related basalt, or enriched MORB, provided you have enough Ba, but that too might be made clear. Perhaps you could also comment on whether the source basalt has to be enriched in minor and trace elements, or not? Does the model not work if the Ba contents in the source basalt are lower? At the moment you appear to want trace element enriched basalt, and I assume they do not reflect smaller degrees of melting in this hotter Earth etc.

It is argued that fractionation, accumulation and assimilation don't change Ba contents significantly. It is difficult to evaluate that statement from the paper, and I wondered why the emphasis was on Ba abundances, rather than suitable trace element ratios, that are arguably more robust as they are less affected by degrees of fractionation etc.? Presumably that might reduce the scatter in the plots of chemical changes through time.

Geochemically, positive Ba anomalies, as in the Stave craton, are clearly interpreted differently from other rocks with similar Ba contents but no positive Ba anomalies. Such distinctions remain important and yet they remain difficult to unpick with the approach as presented.

The conclusion is that only (I assume, hot in this context) subduction zones would provide a sufficiently low geothermal gradient to produce Ba-rich magmas. Perhaps it has been done before, but the key figure for many readers would therefore be how incompatible element contents vary with geothermal gradients, since that appears to be the core message of the approach, and one that is being proposed as a new way forward.

I am undoubtedly biased, but the reduction in the rate of change of crustal growth at $\sim 3\text{Ga}$, which must follow for models in which 70% of the present volume of the continental crust had been generated by that time, is a proxy for the onset of plate tectonics that has stood since Dhuime et al 2012, and I was surprised that it did not warrant a mention.

In summary, the topic is of interest, and new approaches are welcome. However, I am not sure that in its present form this paper comes across as authoritatively as the authors might wish, for the reasons commented on above.

Chris Hawkesworth

Reviewer #3 (Remarks to the Author):

This paper addresses an interesting problem and uses a sound and innovative approach. However, the text is hard to follow and is filled with repetitions. I suggest reorganizing the whole paper. Also, separating Results from the Discussion+Conclusions section may help clarify things.

Detailed comments

Archaean – It's generally spelled 'Archean'

l. 14 (etc.) – not TTG (and not TTGs either) but TTG intrusive suites (or TTG suites) / l. 16 – TTG magmatism, ...

l. 18 – only specialists will understand the reference to a basalt here – you could add the word 'source' to make it clearer, but still, it would be good to explain that TTG are intrusive suites and that their source is a basalt that may melt over a range of P-(T), and this is why you test a range of P-T.

Abstract. In addition to the problem raised in the previous comment, something is missing in the abstract: why is Ba so important? What is the rationale behind focusing on this element? On line 21, we learn that Ba-rich TTG suites is what we need to find – but why? I suggest re-organizing this abstract.

l. 23 – another thing about the abstract is that the aim of the study is not stated clearly. The first part of the abstract suggests the study is about discussing geodynamic setting in general for the Archean. The sentence on line 23 says different – it says the study is about the subduction process and, more specifically, it is about the onset of this process – and is the study about the whole Archean Era or about post-3.0 Ga events? Need to pick one problem and to state it clearly.

l. 24-25. I don't understand this sentence.

Paragraph 1.

TTG are intrusive suites (not rocks)

This paragraph doesn't read well and we are missing information to understand it: TTG a dominant component of the Archean crust, so we can understand to link to geodynamic settings – but why focus on subduction (needs to be explained)? TTG come from the deep partial melting of a basalt – that's not something that can occur only in subduction setting during the Archean.

I. 32 – what does ‘deformation’ has to do with anything here? What is this ‘diverse chemistry’ that is referred to here? What are these diverse models? Besides, chemical heterogeneities within TTG suites has also been interpreted within the framework of a same model – subduction – assuming changing angles for the slab. These remarks of chemical heterogeneity may be better presented elsewhere in the introduction section. Here, we need to understand what the problem is.

Paragraph 2.

L. 44. And metamorphosed – the source is a metamorphic rock. And the next sentence suggests that it can be otherwise (‘For metamorphosed basalt’) – reformulate

I. 45-47 – hard to understand for none specialists. Explain that garnet is a residual phase – that is links to HREE depletion in the melt, ... It’s mostly reformulation. Also, provide the reader with some numbers (what kind of pressure are we talking about here?).

Paragraph 3.

I. 51 – reformulate the beginning of this sentence

I. 51 – that’s new, ‘continental-crust-forming magmatism’ – would be nice to introduce that earlier (that the C.C. was formed by TTG magmatism).

I. 52 – ‘more complex’ – why? What makes you think that? (provide rational behind every assertion)

I. 53 – what are these drips? It’s confusing, unclear what geodynamic setting we are talking about here.

I. 54 – Sentence on Plag. – we have now jumped to another idea and we are lost (need to locate us: here we are talking about lower crust FP, I am guessing anyway).

I. 57 – then differentiation can modify melt X – OK, but how does it relate to the problem at hand? Will it modify the HREE content so much that we won’t be able to know if garnet is a residual phase? The next sentence provides the beginning of an explanation (re-organize).

I. 61 – if this paragraph is about the Sr/Y ratio, then say it in the first sentence (introduction sentence).

Paragraph 4.

I. 64 – before you jump to that, you need to state the problem and the aim: we need a better proxy / then you can say: we will be testing a new one (Ba). And what is petrological modelling doing here? The whole sentence is hard to follow (reformulate)

I. 66-70 – why not say that earlier, where the chemistry of TTG suites is presented? Isn’t this interpreted as the involvement of a mantle component? Anyway, the whole thing should be moved up in the introduction. The whole section on MgO should have its dedicated paragraph and then the problem should be stated clearly.

l. 70-71 – not clear. Modelling cannot generate melt with enough MgO, so fractionation must be limited? What is the link, what is fractionation doing here if we cannot even reproduce the primitive melt (such experiments tell us nothing on the extent of fractionation).

l. 73 – yes, mantle involvement, you are not the first to suggest that (see work of Martin for example).

l. 76 – what mass balance calculation? Need reference and explanation.

l. 77 – where does this come from?

l. 78 – what are sediments doing here now? / l. 82 – and there goes sediments. It is very confusing – what model are we testing here? Subduction? Then start from subduction, from what we can expect in such setting, and discuss TTG chemistry by always stating what it means for the subduction model.

Idem for the rest of the paragraph, need reorganizing. And all remarks need to be explained and referenced (e.g., l. 86 – Biotite accumulation is unlikely – based on what evidence?).

This section is very confusing because it concludes that Ba is the best proxy: then work done, let's conclude the paper. You need to reorganize all this and to orient the paper toward the conclusion that is stated here: Ba is best because, according to our Results, it is a proxy to the composition of the primitive melt (that's for the Discussion section, not the Introduction section).

L. 90 – referring to the EAT may be done later – you need to state where this composition comes from (reference).

Paragraph 5.

Line 96. Ba content of what? I don't expect the whole rock Ba composition to change much during metamorphism.

l. 100. Would be nice to start with that (what are we trying to model?). Then move to how the model is performed, and describe the results.

l. 104. 'hot subduction' – might be best to refer to model A, B, ... - or start the paragraph like this: model A has these characteristics and aims at modelling the hot subduction scenario, ... (reorganizing).

l. 110 - ... according to tectonic setting. You should separate things: first talk about modelling and its results, and later in the text relate results to geological realities.

Paragraph 6.

l. 114 – I am confused, I thought only one protolith composition (EAT) was tested? Do you mean water-content?

Paragraph 7.

The first sentence has nothing to do here – it doesn't do a good job introducing a paragraph that describes primitive melt composition. Focus this paragraph on describing Results.

l. 136-etc. – need to cover that earlier in the text. A section dedicated to presenting the EAT source and discussing Ba distribution in Archean basalts and TTG would be useful – to present before presenting the Results of the model.

Paragraph 8.

This is not located in the right place. You need to dedicate part of your text (and only one part of it) to the different possible geodynamic scenarios. Idem for line 157 – that is the fourth time we get back on this fractionation issue – you need to discuss fractionation once and for all. The whole text needs reorganizing.

Paragraph 9.

l. 162. Not clear that Ba is now establish as a proxy for subduction (not tectonic setting according to what I read before).

See remark made earlier. It would be more pertinent to first present what Ba looks like in these rocks, and to then move on to modelling, rather than doing it the other way round.

Would be nice to say a word on the dataset – are we looking at a global compilation? Done by who?

Paragraph 10.

l. 170. ‘indicates their formation in an intraplate setting’ – according to your model. This is a major flaw of this paper: too many affirmation that are lacking context or references. And what does ‘intraplate setting’ looks like for Archean TTG?

l. 173. ‘become complicated’ – reformulate / ‘change point’ or ‘turning point’ (there is a GSA volume that should come out soon that discusses the ‘turning point’ concept for the North American craton).

. 175 – why ‘delayed’? They are different (diachronicity)

. 176. The TTG suites of the Slave craton (not the Slave craton)

. 178-179 – reformulate (and don’t repeat ‘Slave craton’ so many times)

. 180 – useless sentence (it is what you suggest in the whole section)

. 181. Remove ‘been’

. 184. We are going back on this fractionation things (fifth time) – reorganize

. 188. Stagnant-lid vs plate tectonic are notions to be introduced much earlier in the text

l. 190-194. Here you are saying that other data (like mantle cooling) are useless so we need a new proxy – this should be said in the Introduction. Here, most of the paragraph talks about diachronicity – mantle cooling is global and cannot explain diachronic event. So what can explain this diachronism?

I. 195-197. These 2 sentences make no sense: if the oldest rocks record the shift, then there are no older rocks to record low Ba content. And what is 'rock preservation' doing here? If there is no older TTG suite, could be they never existed (doesn't necessarily mean they got eroded).

Paragraphs 10 + 11 + 12

I don't see the plus-value of these data, especially at this stage of the demonstration. These are statements for the Introduction. Also, given that Palin et al. (2020) recently reviewed many such evidences, I would rather shorten and move this paragraph to the Introduction – alternatively, just delete it.

Idem for paragraphs 11 and 12 – this sounds like an Introduction.

Figure 4. Figure 4a is very important to the text, but it is unreadable – it is hard to see what you mean by change-point and which data correspond to which craton. Need to revise the design of this Figure.

Figure 5. Useless Figure (this has been published)

Responses in blue.

“Revisions in red”.

Line numbers refer to manuscript version *without* tracked changes.

Reviewer #1 (Remarks to the Author):

Review of Nat. Comms manuscript “Barium content of Archaean continental crust . . .” (by Guangyu Huang et al)

The thought of using Ba in this way is quite an interesting a novel approach to a significant problem in how we view the (various and changing) origins of TTG and hence, earlier crustal evolution. There is a lot to like about this study and I certainly think it is a topic suitable for Nat. comms. The manuscript is generally quite well written but there are a few areas where arguments become a bit muddled – and where all of the listed co-authors perhaps could add an additional polish. I’m not convinced that the interpretation presented here is the only way to interpret that data presented, but it is new, certainly possible and will create interest to all struggling with ideas about how and when early crust forming processes operated. I would certainly like to see the idea put out there for wider consideration, after a bit of clarity on a few points and recommend publication with moderate revision.

One point that I have difficulty with is suggesting that a decrease in the LILE content of TTG at a point where subduction is suggested to become their new mode of occurrence is somewhat counter intuitive. You can’t attribute this all to a flat-subduction regime because you then deny the evidence for metasomatic enrichment in the mantle source of some Meso- to Neoproterozoic mafic sequences. **Reviewer #1 subsequently clarified** “I simply meant that one of the characteristics of subduction magmatism is the high LILE content so it seems odd that you should define the onset of subduction by a decrease in LILE.”

Thanks for the clarification from reviewer 1. In fact, we indeed use the positive shift of the LILE (specifically Ba) to recognize the onset of subduction. It might be our fault if we hadn’t made this clear enough. We have added mention of the “Ba positive shift” in the Abstract and anywhere else where the matter need to be clarified thereafter.

“The Ba contents in TTG suites show a statistically significant positive shift at 3.7 Ga and do not show an additional step change thereafter, implying an irreversible change took place in global geodynamics (Fig. 4a).”

Also, you are putting a lot of trust in an element that is notoriously mobile, in TTG forming discernible correlation with basically nothing . . . except, in some cases, for Sr! The fact that we see change points, and of a systematic direction itself is surprising and possibly quite meaningful.

Thanks for your suggestion. We have plot the secular change of Sr/Y of the TTG suites. As you could see in the following figure, the Sr/Y vs age plots of

the TTG suites are noisy, and do not show better results than Ba content. We take this comment seriously, as many readers might want to see this plot. Therefore, we add this figure as Supplementary Figure 7.

Perhaps related to inherent Ba scatter - hopefully another reviewer is better able than I am to judge the validity of the statistics, but many of the change points shown in the sup data seem to be very tenuous. There does nevertheless appear to be a trend consistent with the given interpretation.

We have edited the Figure 4a. The positive shifts are consistent with those in Supplementary Figure 5 in the revised version.

L22-23 - this is a bit of a nonsense - how can we have regional global subduction?

In fact, what we want to express is both the temporal and spatial patterns of the onset of subduction in early Earth history. Based on our study, we found that the age of the initiation of subduction is diachronous for different cratons. Spatially, then, the onset of subduction would therefore evolve from regional (e.g., Slave craton, North Atlantic craton) to global over time.

“We then identify statistically significant increases in the Ba contents of TTG suites worldwide as recording the diachronous onset of subduction from regional at 4 Ga to globally complete after 3 Ga. ”

L32 – not really much ‘grey gneiss’ in any of the Australian Archean regions. Sorry for not knowing this. We just learn this from Moyer’s paper, as we have cited. Acknowledging this misrepresentation, we have deleted the word “grey”. “Most TTG suites in Archean cratons occur as essentially identical deformed gneisses and have consistent and simple mineralogy, but their diverse compositional characteristics (e.g., Sr/Y, La/Yb, and Nb/Ta) has led to a variety of models for how Earth’s earliest continental crust formed and subsequently evolved through time. ”

L60-62 – I certainly agree that this seems to true.

L72-73 - I seriously doubt that this is the case. I can accept that some Archean high Mg andesitic hybrids might form through interaction of small amounts of felsic material with much larger volumes of mafic or ultramafic magma (e.g. Barnes and Van Kranendonk) but physical mixes of mafic magma into melts as felsic as the >70wt% silica melts you are talking about typically manifests themselves in the field very, very, clearly for what they are. Removing these from your dataset basically ensures that your argument about a lack of evidence for fractionation becomes a self-fulfilling prophesy. You might like to consider an alternative – that these higher Mg# samples have a more sanukitoid-like petrogenesis. In fact, some researchers have already proposed the possibility of magma mixing between felsic melts (e.g., TTG magma) and high Mg rocks, as suggested by reviewer #3 for us to add Martin’s references. However, we do agree with the alternative opinion (i.e., sanukitoid-like petrogenesis). We have added this alternative petrogenetic model and the citation (Smithies and Champion, 2000).

Martin H, Moyer JF. (2002) Secular changes in tonalite-trondjemite-granodiorite composition as markers of the progressive cooling of Earth. Geology 30, 319–322.

Smithies RH, Champion DC (2000) The Archean high-mg diorite suite: links to tonalite-trondjemite-granodiorite magmatism and implications for early archaean crustal growth. J Petrol 41:1653–1671.

“Some MgO-rich melts or rocks (e.g., komatiite, picrite, and peridotite) may need to be assimilated into a TTG magma or into the source rock to produce the observed compositions^{16–17}. ”

L 77-79 - this might be the case, but the further we go back in the Archean, the more ‘sodic’ systems become. If subduction was occurring in the early Archean, would

we necessarily expect the same LILE (and particularly, K, Rb) enrichments then as we see today? Certainly more evidence exists in the Neoproterozoic, but this is when Ba concentrations in TTG apparently decrease!!

It is true. If the magmas of early Earth were more “sodic”, then we do not need to worry about the assimilation of sediments, since the sediments would also be TTG-like in composition. The high Ba content must come from the melt composition itself, as no other high Ba source would enhance the Ba content in TTG magma. What we want to do is to demonstrate that the high Ba content in TTG magma would only have resulted from their initial composition.

L87-88 - this apparent negative correlation between Ni and Ba is certainly not at all clear to me from this figure.

It is our fault that we didn't describe it properly. What we want to express is that high-Ba TTG usually have low Ni content. We have made the correction. “Importantly, Ni is also compatible in biotite, but Ni contents in most high-Ba TTG suites (Ba > 1,000 ppm) does not exceed 30 ppm, which is even lower than those of low-Ba TTG suites (Fig. 1d).”

L96-98 - after quite a reasonable review of currently accepted views on TTG formation, it seems a bit surprising that water-saturated melting is now the sole consideration? This probably needs a line or two of justification!

Thanks! We have added some description for why we just model water-saturated melting.

“According to the assumption that fluid-absent melting is the main mechanism to generate TTG-like melts¹⁵, previous work has explored the evolution of melt composition under water-unsaturated conditions.”

L114 - I accept that we have to make such assumptions, but there are several Moyen papers that clearly articulate exactly how important source composition is.

It is true. The source composition is important for the melt composition, as we described before stating our assumption. It has been indeed articulated by Moyen's papers. We have added the citation to further bolster the basis for this assumption.

Moyen JF, Martin H (2012) Forty years of TTG research. Lithos 148:312–336.

L137 - . 160 ppm seem very low for TTG. The average Ba in 229 Pilbara TTG meeting the definition given above is 560ppm and for 1011 samples from the Yilgarn meeting these criteria Ba = 650ppm. I certainly agree that there is not enough variability in the primary Ba content of Archean basalt to account for the Ba contents

of TTG and suggest that in many cases the source is clearly not that simple. I think this is a caveat you need to stress.

It is our fault that we made a mistake here. The 160 ppm value is for basaltic composition, not TTG. We have made the correction.

“The average documented Ba content of enriched Archaean tholeiitic basalts is 107 ppm, and most have less than 160 ppm, indicating that there is not enough variability in the primary Ba contents of Archaean basalts to account for the diversity of Ba contents in TTG suites, especially in the early Archaean when general compositional diversity (e.g., in Ba) is predicted to have been lower²⁶.”

L139-140 – if this is indeed the case then the findings might be more relevant to the earlier Archean, when general compositional diversity (e.g. in Ba) is predicted to have been lower (reduces the issues of source variations).

Thanks. As reviewer #1 suggested, we do want to express that the source variations in the early Archaean is limited, especially in Ba.

L154 – I’m not sure that all of Mike Brown’s data was restricted to the abovementioned regions.

In fact, Mike Brown’s data include the data of the regions that we mentioned.

L155-156 – not a sentence – and why would we expect a transition in Sr/Y through time related to plagioclase accumulation anyway?

In previous study, the Sr/Y ratio has been widely used to distinguish HP TTG from other types of TTG suites. If these HP TTG suites with high Sr/Y would represent magma generation depth, then we would expect a transition of Sr/Y with age once subduction initiated. However, such a transition was not identified. This is likely because the Sr/Y of TTG magma would be influenced not only by the initial melt, but plagioclase accumulation. What we want to express is that Sr/Y would not be a good proxy for identifying this transition. Based on our study, we find that TTG magma generated in a subduction setting would have high Ba content when the partial melting degree is lower than 10 vol.%, and high Ba content would not be influenced by magma assimilation and fractional crystallization. High Ba content in TTG suites would be a new finding and represent a robust proxy for ancient subduction settings.

L169-170 – using ‘globally’ here is very misleading given n=1 or 2. You have no clue that its global.

Thanks. It is true. We have deleted the word ‘globally’.

“The Ba contents in TTG suites show a statistically significant positive shift at 3.7 Ga and do not show an additional step change thereafter, implying an irreversible change took place in global geodynamics (Fig. 4a).”

L174 - 175 - if the change is relevant only to the North Atlantic craton then don't refer to it as a global trend in the first place. This makes all of this paragraph very confusing. There is a very important and valid point to be made here, but it is being muddled.

As suggested by both reviewers #1 and #3, this paragraph might be difficult to follow for readers. Therefore, we have rewritten. If we just use global TTG data to identify the Ba shift, then the age of 3.7 Ga would be considered as the onset of subduction. However, these data should be further explored spatially. For example, we should not take the early Ba shift of the North Atlantic craton to represent that of the global Earth. As we present in Supplementary Figure 5, the age of the Ba shifts of the various cratons are different, with NAC at 3.7 Ga, Baltica and Pilbara at 3.5 Ga, Dharwar at 3.4 etc. The revised text should be more clear.

“Before 3.7 Ga, with the sole exception of Slave craton, nearly all TTG suites preserved in the rock record have low Ba contents, which indicates their formation in an Archaean drip setting. The Ba contents in TTG suites show a statistically significant positive shift at 3.7 Ga and do not show an additional step change thereafter, implying an irreversible change took place in global geodynamics (Fig. 4a). Upon closer inspection, the ages of the positive shift in Ba become complicated when plotted for specific cratons (Fig. 4a; Supplementary Fig. 5). The apparent “global” 3.7 Ga positive shift of all TTG suites is in verity likely a signature of the North Atlantic craton, while the ages of shifts in other cratons are typically younger and highly diachronous (Fig. 4a). The TTG suites of the Slave craton preserve high-Ba values as early as 4.0 Ga, which it maintains throughout the rest of the Archaean. A strongly positive Ba anomaly has been identified at 3.5 Ga in Slave (Fig. 4a), which, if interpreted as an indicator of subduction, is independently corroborated by a seismically imaged slab beneath the Slave craton dated to ca. 3.5 Ga (ref. ⁶¹). The ages of the various positive shift within different cratons might result from different starting times of subduction. Subduction might have occurred in the Slave craton since 4.0 Ga, followed by initiation at 3.7 Ga in the North Atlantic craton, while most others initiated at 3.2–3.0 Ga (e.g., Baltica, Amazonia, and Pilbara), or even much later at 2.7 Ga (Superior craton). ”

L180-183 - so you are clearly NOT talking here about global subduction - this point needs to be made.

In fact, we are talking about global subduction. However, we could not see the onset of subduction as a whole. The initiation of subduction would be influenced by a lot of factors (e.g., mantle temperature, radiogenic heat production, density contrast between the lithospheric mantle and the convective mantle, lithospheric thickness, etc.). Furthermore, Earth would also not be homogeneous in all aforementioned factors, so subduction should

likely only be initiated at a location where all factors are suitable, which should naturally only be regional in scale in the beginning. With subduction have been initiated in more and more regions, subduction would evolve to become global in scale eventually, as has been previously argued for by 2 Ga at least (Wan et al., 2020). By using the Ba-in-TTG proxy for subduction initiation, we could recognize the order of these regions, which was listed in this sentence. Therefore, we are talking about the *process and evolution towards* global subduction.

“What accounts for the observed diachroneity? The initiation of the subduction would be a function of several factors, including mantle temperature, radiogenic heat production, density contrast between the lithospheric mantle and the convective mantle, and lithospheric thickness⁶². The whole Earth would not be homogenous in all these factors, so we may not expect the onset of subduction to be a globally isochronous event as has often been portrayed in past work. Subduction likely is initiated once all these factors are suitable, which should naturally only be regional in scale at the beginning, and eventually evolve to a global scale once other regions are suitable⁶³. Based on the notable change in the Ba content of TTG suites identified here, we instead propose a subduction propagation model, in which a transition from a stagnant-lid regime to plate tectonics would have occurred from regional (e.g., Slave and North Atlantic cratons) to global in scale. Such an age for the onset of plate tectonics is consistent with Earth’s thermal history according to modelling and petrological data (Fig. 5). Because of uncertainties in observational inferences as well as theoretical models, the cooling history of Earth’s mantle cannot tightly constrain when plate tectonics might have emerged from stagnant lid convection (a putative pre-plate-tectonics regime) (Fig. 5), thus rendering the newly developed Ba proxy for subduction a critical constraint. Furthermore, the ages of Ba positive shifts of various cratons correlate strongly with the ages of the oldest rocks of each craton (Fig. 4b). According to our proposal that Ba positive shifts indicate subduction initiation, this correlation would thus imply an important relationship between rock preservation and subsequent subduction.”

L183-187 - this is a misleading misinterpretation of what is said in REF11 - which in fact claims that at least some HP TTG might result through fractional crystallisation of primitive sanukitoid magmas - and proceeds to demonstrate this using a large dataset of temporally and spatially related samples that define a complete liquid line of descent!

Sorry for this. Considering the comment of Reviewer #1 and #3, we have moved this sentence to where fractional crystallization first appears. We have edit our misinterpretation.

“Appreciable amounts of plagioclase accumulation and fractional crystallization of hornblende would induce HREE depletion and enhance Sr/Y in the melt, similar to that of HP TTG magma ^{32–35}. ”

203-204 – as it still is with the Ba proxy! i.e somewhere between 4 and 2.7Ga. . . or after!

In fact, we have explained the reason for the contentiousness of previous proxies. Most of these proxies either record the first appearance of specific rock types or metamorphic facies or indirectly infer the nature of the continental crust. These ages should be considered as the upper limit age of plate tectonics. Our work shows the Ba shift in TTG suites would be a robust proxy for the onset of subduction. In other words, we believe that the composition shifts of TTG would be always more robust than the first appearance of specific rocks (such as blueschist). We do not know whether older blueschists exist. The age of the onset of subduction among different cratons would be different as the lithospheric factors are various, as discussed above. Therefore, although the onset ages between 4 and 2.7 Ga is a range, it is robust and meaningful. If the reviewer #1 indeed asked for the first appearance of subduction in the earth history, it should be happened at the Slave craton at 4.0 Ga. While we understand the reviewer’s apt criticism (as it relates to previous studies), we hope they appreciate that we are careful to explore the spatiotemporal patterns of our proxy to better understand the wide range in ages of the Ba shifts: it’s not an ambiguous proxy, rather the process it records is simply diachronous throughout the world.

L217 WHY do we need to search for a specific date!!!

It is a rather good question, but as the reviewer is well aware, framing the onset debate this way is the nature of the current discussion on the topic. We hope our work can help the discussion progress in the future to better reflect the reviewer’s perspective.

This (the claimed pervasive nature of the required process – L185) does, however, highlight a particular point of relevance – that indeed perhaps supports the FC model for HP TTG rather than the one presented here – and that is that HP TTG (even using Ba concentrations as a proxy) are in fact not common, and in some cratons almost absent. For something that’s heralding such a significant change in Earth dynamics, perhaps we would expect to see more of it preserved.

With all due respect to the reviewer that may have their own opinion, we want to avoid talking about the FC model as, in our opinion, it is unlikely to be the main model for TTG generation globally. If fractional crystallization of hornblende from sanukitoid magma would generate TTG suites, it would mostly happen after 3.0 Ga. As far as we know, sanukitoids with ages older than 3.0 Ga are scarce, and are only found in eastern Pilbara craton. We could not see one exception as a common model.

190-191 “Such an age for the onset. . . .” What age? – you have given an age range, so where within that range are you putting the ‘onset’ – there is no real ‘onset’ and in all of this there is no indication of anything that clearly resembles a Plate tectonic regime of globally connected destructive margins supported by ridges.

As discussed above, we are talking about the onset of subduction temporally and spatially. As Earth is not homogenous, the onset of subduction likely varies spatiotemporally. Figure 4 illustrates the order of the onset of subduction for various cratons.

Reviewer #2 (Remarks to the Author):

This paper tackles the topical subject of the conditions under which Archaean TTG were generated using petrogenetic modeling to estimate the Ba contents of melts generated under different thermal conditions. The minerals stable in the preferred basalt composition, and involved in partial melting, are sensitive to the geothermal gradients, and it is concluded that suitable Ba contents are generated at relatively low geothermal gradients. In some ways this is a confusing paper in that Ba contents are highlighted, and yet it is the thermal gradients that are key to the conclusion. That point has been made before by Tim Johnson and colleagues in Nature, and then, understandably, the key conclusions are expressed through trace element abundances. While Ba appears to work well, it is not clear whether there is something distinctive about Ba in this context, or other incompatible elements would have worked equally well. Th is often used as it may be less mobile during alteration and metamorphism.

Thanks. We have modeled the evolution of Th composition in melts during partial melting, which the reviewer can find below. Th is strongly compatible in epidote ($K_{d_{Th}}=156$, Bédard, 2006), which is usually stable at temperatures below 750 °C and pressures above 1.0 GPa. In a hot subduction setting, we could see rather low Th content in modeled melt when melt proportion lower than 7 vol.%. Ba is obviously high at the same time. We then could expect an apparently high Ba/Th ratio at the beginning of partial melting in a subduction setting. However, biotite shares the same stable temperature field with epidote. As a result, these incipient melts are usually granitic, but not TTG-like, in composition. When melting degree exceeds 7 vol.% in the hot subduction setting, the melt would be TTG-like in composition, but the Th content and Ba/Th would be in the same range with those in the intraplate setting. Meanwhile, we found a lot of TTG compositions are lacking Th composition in the dataset of Moyen and Martin (2011) and Johnson et al. (2019). As a result, the statistical analyses of Th or Ba/Th composition vs age plots would be rather sparse and statistically ambivalent. As we have describe in the manuscript and in response to reviewer #1, the Ba content in TTG might be altered during magma assimilation and fractional crystallization, but

no process would enhance the Ba content. Later stage metasomatism and metamorphism might alter the Ba content in TTG, but other LILE (e.g., K, Rb, Sr) were not found to be positively correlated with Ba content. Ba content would therefore be mostly derived from the original magma compositions.

Bédard JH (2006) A catalytic delamination-driven model for coupled genesis of Archaean crust and sub-continental lithospheric mantle. *Geochim Cosmochim Acta* 70:1188–1214.

Moyen JF (2011) The composite Archaean grey gneisses: petrological significance, and evidence for a non-unique setting for Archaean crustal growth. *Lithos* 123:21–36.

Johnson TE, Kirkland CL, Gardiner NJ, Brown M, Smithies RH, Santosh M (2019) Secular change in TTG compositions: Implications for the evolution of Archaean geodynamics. *Earth Planet Sci Lett* 505:65–75.

Johnson et al used a broadly similar approach to argue that the first continents were not formed by subduction. It is unclear whether you are in agreement given the low Ba contents of the older TTG, or in disagreement in ways that are not clear to the reader, but it would be a great help if that was addressed.

Our modeling result shows that the hot subduction setting would generate high-Ba TTG suites, while the drip environment would not. Ba content would not be enhanced by other magmatic process such as magma assimilation and fractional crystallization. Therefore, Ba content should be a suitable proxy to test the argument of Johnson et al. (2017). Our results show that the oldest TTG suites in most old cratons were not generated by hot subduction, except for the Slave craton. Subduction might have existed in the Slave craton since 4.0 Ga. However, it still needs to be tested in future work, as rocks with ages of ca. 4.0 Ga are rather rare.

Johnson TE, Brown M, Gardiner NJ, Kirkland CL, Smithies RH (2017) Earth's first stable continents did not form by subduction. *Nature* 543:239–242.

It is striking that only two major geological settings were selected. The conclusion is that one works well, but that is not the same as saying it is the preferred solution given all the settings that might have been present in the Archaean – which is how the paper might be interpreted. Can it be shown that cold subduction can also be ruled out with this approach?

Thanks. Sure. We have modeled a phase diagram, with pressure ranging from 2.05 to 3.25 GPa and temperature ranging from 600 to 1000 °C. A thermal gradient of 250 °C/GPa was labeled with a bold white line on the P - T diagram. The thermal gradient of cold subduction could be found in Hasebe et al. (1970) and Martin (1986). As the reviewers can see, the thermal gradient of 250 °C/GPa is almost parallel to the isopleths of the melt proportion. As a result, the generated melt proportion (less than 4 vol.%), which is lower than the “melt connectivity transition” of 7 vol.% (Rosenberg and Handy, 2005), would not be enough to extract from the protolith. Therefore, the partial melting of a subducted slab would not be able to generate TTG magma, and could be ruled out by this approach. We have added a description of this result. References are properly cited, and the new P - T diagram can be found as Supplementary Figure 6 and below.

Hasebe, K, Fujii, N, Uyeda, S. (1970) Thermal processes under island arcs: Tectonophysics, 10: 335-355.

Martin, H. (1986) Effect of steeper Archean geothermal gradient on geochemistry of subduction-zone magmas. Geology, 14: 753-756.

Rosenberg CL, Handy MR (2005) Experimental deformation of partially melted granite revisited: implications for the continental crust. J Metamorph Geol 23:19–28

“As an additional possibility, a geothermal gradient of $250\text{ }^{\circ}\text{C GPa}^{-1}$, corresponding to a cold subduction setting^{39,42}, was also explored. For a cold subduction setting, calculated melt proportion isopleths lie almost parallel to the geothermal gradient, leading to no more than 4 vol.% of melt generation (Supplementary Fig. 6), which would also not be able to be extracted from the protolith⁴³. Such results can thus rule out cold subduction for TTG magma generation.”

I am not sure of the source of the thermal conditions for hot subduction and drip tectonics, as references 33 and 34 are relatively geological. Yet the thermal conditions are key to the arguments presented, and how robust are the geothermal gradient estimates for the two settings?

Sorry for this. The geothermal gradients of the hot subduction setting can be found in Martin (1986) and Brown and Johnson (2018). The thermal gradient of drip tectonics could be found in Moyen and Martin (2012) and Sizova et al. (2018). The thermal gradient of the cold subduction setting can be found in Martin (1986) and Brown and Johnson (2018). We have added these references.

Redacted

Martin H (1986) Effect of steeper Archean geothermal gradient on geochemistry of subduction-zone magmas. Geology 14: 753–756
Brown M, Johnson T. (2018) Secular change in metamorphism and the onset of global plate tectonics. American Mineralogist 103: 181–196
Moyen JF, Martin H (2012) Forty years of TTG research. Lithos 148:312–336
Sizova E, Gerya T, Brown M. Stüwe K. (2018) What Drives metamorphism in early Archean greenstone belts? Insights from numerical modeling. Tectonophysics 746: 587–601

I am sorry if I missed it, but it was difficult to find the minor and trace element pattern of the source basalt, and that would certainly help the geochemists. I suspect you might argue that it doesn't matter whether it is a subduction-related basalt, or enriched MORB, provided you have enough Ba, but that too might be made clear. Perhaps you could also comment on whether the source basalt has to be enriched in minor and trace elements, or not? Does the model not work if the Ba contents in the source basalt are lower? At the moment you appear to want trace element enriched basalt, and I assume they do not reflect smaller degrees of melting in this hotter Earth etc.

Thanks. We are agreed with you, and made a description of the modeled source composition. It is a composition we cite from Ge et al. (2019), and we have added the reference. They sorted the composition from GOEROC

dataset, and sorted the Archaean moderate enriched basaltic compositions. These compositions have Th/Nb ratios above 0.1. As they describe, these compositions are suitable for generating large volumes of TTG magma. Other compositions (such as MORB, OPB, OIB) are either depleted in LILEs, or rare in nature, and are not able to account for a large volume of TTG generation. The higher Th/Nb (>0.1) would not be necessarily related to the subduction setting. As described and discussed in Moyen and Laurent (2018) and Smithies et al., (2018), the oblique array joining a rather depleted portion of the mantle array to the arc field can result from the assimilation of continental crust.

Moyen JF, Laurent O. (2018) Archean tectonic systems: A review from igneous rocks. Lithos 302-303: 99-125

Smithies RH, Ivanic TJ, Lowrey JR, Morris PA, Barnes SJ, Wyche S, Lu YJ. (2018) Two distinct origins for Archean greenstone belts. Earth and Planetary Science Letters. 487: 106-116

“It has been discussed that TTG suites are highly enriched in large-ion lithophile elements (LILEs; e.g., K, Rb, and Ba), and cannot be generated from mid-ocean ridge basalts (MORB) and oceanic plateau basalts (OPB), which are depleted or not enriched enough in LILEs^{26,37}. Oceanic island basalts (OIB), even though they are enriched in LILEs²⁶, also cannot generate large volumes of TTG magma owing to their restricted abundance. Ge et al. ²⁶ compiled more than 1,000 enriched Archaean tholeiitic basalts that are characterized by moderately enriched LILE and have Th/Nb > 0.1 and suggested these basalts would be a suitable source for generating large volumes of TTG magmas. These basalts have a wide range of trace element compositions, and would not necessarily be related to subduction since the oblique array joining a rather depleted portion of the mantle array to the arc field would result from continental crust assimilation³⁸. To verify the hypothesis that the diversity of original melt compositions accounts for high-Ba TTG magmas, the average enriched Archaean tholeiitic basalt composition²⁶ was used to carry out petrological modelling at different geothermal gradients and water contents, which allowed predictive modelling of melt compositions in different geodynamic settings. The average documented Ba content of enriched Archaean tholeiitic basalts is 107 ppm, and most have less than 160 ppm, indicating that there is not enough variability in the primary Ba contents of Archaean basalts to account for the diversity of Ba contents in TTG suites, especially in the early Archaean when general compositional diversity (e.g., in Ba) is predicted to have been lower²⁶.”

It is argued that fractionation, accumulation and assimilation don't change Ba contents significantly. It is difficult to evaluate that statement from the paper, and I wondered why the emphasis was on Ba abundances, rather than suitable trace

element ratios, that are arguably more robust as they are less affected by degrees of fractionation etc.? Presumably that might reduce the scatter in the plots of chemical changes through time.

Thanks for your suggestion. We have modeled the secular changes of Sr/Y and Ba/Th in TTG suites. As you could see in the following figure, neither show better results than Ba content. The plots are noisy. We believe that readers might want to see the Sr/Y vs age plot, so we added the Sr/Y plots as Supplementary Figure 7.

Geochemically, positive Ba anomalies, as in the Stave craton, are clearly interpreted differently from other rocks with similar Ba contents but no positive Ba anomalies.

Such distinctions remain important and yet they remain difficult to unpick with the approach as presented.

Slave craton is a little bit different as we discussed above. Our results show that the oldest TTG suites in most old cratons were not generated by hot subduction, except for the Slave craton. Subduction might have existed in the Slave craton since 4.0 Ga. However, it still needs to be tested in future work, as rocks with ages of ca. 4.0 Ga are rather rare. In some other cratons, such as NAC, Pilbara craton, etc., the Ba contents of some the oldest TTG could also be high ($\log(\text{Ba}) = 3$, i.e. $\text{Ba} > 1,000$). However, these data are outliers, and we cannot base our statistically-based conclusion on outliers.

The conclusion is that only (I assume, hot in this context) subduction zones would provide a sufficiently low geothermal gradient to produce Ba-rich magmas. Perhaps it has been done before, but the key figure for many readers would therefore be how incompatible element contents vary with geothermal gradients, since that appears to be the core message of the approach, and one that is being proposed as a new way forward.

Figures 1, 2, and 3 depict Ba content variability with geothermal gradients. We use these figure to verify that high-Ba TTG suites would be a proxy for identifying the presence/onset of subduction. To our knowledge, this has not been done before. Figure 4 presents the Ba shift for TTG suites globally throughout the Archaean.

I am undoubtedly biased, but the reduction in the rate of change of crustal growth at $\sim 3\text{Ga}$, which must follow for models in which 70% of the present volume of the continental crust had been generated by that time, is a proxy for the onset of plate tectonics that has stood since Dhuime et al 2012, and I was surprised that it did not warrant a mention.

Sure. We have cited the reference of Dhuime et al. (2012).

“Exploring the rate of change of crustal growth, the petrogenesis of TTG magmas, and secular changes in TTG compositions may provide insights into the geodynamic settings of their formation and thus, potentially constrain the age of the onset of global subduction²⁻¹²”

In summary, the topic is of interest, and new approaches are welcome. However, I am not sure that in its present form this paper comes across as authoritatively as the authors might wish, for the reasons commented on above.

Chris Hawkesworth

Reviewer #3 (Remarks to the Author):

This paper addresses an interesting problem and uses a sound and innovative approach. However, the text is hard to follow and is filled with repetitions. I suggest reorganizing the whole paper. Also, separating Results from the Discussion+Conclusions section may help clarify things.

Detailed comments

Archaean – It's generally spelled 'Archean'

As Nature Communications is published in the UK, we employ the UK spelling "Archaean", as is common praxis.

l. 14 (etc.) – not TTG (and not TTGs either) but TTG intrusive suites (or TTG suites)
/l. 16 – TTG magmatism, . . .

We have change all the TTG and TTGs to TTG suites.

l. 18 – only specialists will understand the reference to a basalt here – you could add the word 'source' to make it clearer, but still, it would be good to explain that TTG are intrusive suites and that their source is a basalt that may melt over a range of P-(T), and this is why you test a range of P-T.

Good distinction. We have added the word 'source'.

"We conduct petrological modelling over a range of pressure–temperature conditions relevant to the Archaean geothermal gradient using an average enriched Archaean basaltic source composition to predict Ba concentrations in TTG suites, which is difficult to enhance after magma generation."

Abstract. In addition to the problem raised in the previous comment, something is missing in the abstract: why is Ba so important? What is the rationale behind focusing on this element? On line 21, we learn that Ba-rich TTG suites is what we need to find – but why? I suggest re-organizing this abstract.

We have re-organized the Abstract. The rationale for focusing on Ba is stated.

"We conduct petrological modelling over a range of pressure–temperature conditions relevant to the Archaean geothermal gradient using an average enriched Archaean basaltic source composition to predict Ba concentrations in TTG suites, which is difficult to enhance after magma generation. "

l. 23 – another thing about the abstract is that the aim of the study is not stated clearly. The first part of the abstract suggests the study is about discussing geodynamic setting in general for the Archean. The sentence on line 23 says different – it says the study is about the subduction process and, more specifically, it is about the onset of this process – and is the study about the whole Archean Era or about post-3.0 Ga events? Need to pick one problem and to state it clearly. We have edited this sentence to make our primary focus more clear.

“The results indicate only low geothermal gradients corresponding to hot subduction zones ($\sim 450\text{ }^{\circ}\text{C GPa}^{-1}$) would produce Ba-rich TTG suites, thus representing a proxy for the onset of subduction. ”

L. 24-25. I don't understand this sentence.

We have deleted this sentence.

Paragraph 1.

TTG are intrusive suites (not rocks)

Thanks. We have changed TTG rocks to TTG suites.

This paragraph doesn't read well and we are missing information to understand it: TTG a dominant component of the Archean crust, so we can understand to link to geodynamic settings – but why focus on subduction (needs to be explained)? TTG come from the deep partial melting of a basalt – that's not something that can occur only in subduction setting during the Archean.

As reviewer #3 suggested, we have reorganized this paragraph. “Tonalite–trondhjemite–granodiorite (TTG) suites and their metamorphosed equivalents (TTG gneisses) are the dominant components of Archean continental crust¹⁻². TTG suites are typically interpreted as the product of partial melting of a hydrated basaltic source³. Most TTG suites in Archean cratons occur as essentially identical deformed gneisses and have consistent and simple mineralogy, but their diverse compositional characteristics (e.g., Sr/Y, La/Yb, and Nb/Ta) has led to a variety of models for how Earth's earliest continental crust formed and subsequently evolved through time. Among these continental-crust-forming magmatic models, partial melting of subducted slabs or the lower portions of thickened oceanic crust are the most widely cited mechanisms for the generation of TTG magma⁴⁻⁵. Exploring the rate of change of crustal growth, the petrogenesis of TTG magmas, and secular changes in TTG compositions may provide insights into the geodynamic settings of their formation and thus, potentially constrain the age of the onset of global subduction²⁻¹².”

l. 32 – what does ‘deformation’ has to do with anything here? What is this ‘diverse chemistry’ that is referred to here? What are these diverse models? Besides, chemical heterogeneities within TTG suites has also been interpreted within the framework of a same model – subduction – assuming changing angles for the slab. These remarks of chemical heterogeneity may be better presented elsewhere in the introduction section. Here, we need to understand what the problem is.

Thanks. In terms of deformation state, we just want to express the similarity of these TTG suites. For the diverse chemistry, we have added the trace element ratio (Sr/Y, La/Yb, Nb/Ta etc.). For the diverse modeling results, we

have described them in this paragraph: they are for the partial melting of a subducted slab and the lower part of the thickened oceanic crust. The revisions can be seen immediately above.

Paragraph 2.

L. 44. And metamorphosed – the source is a metamorphic rock. And the next sentence suggests that it can be otherwise (‘For metamorphosed basalt’) – reformulate

Thanks. We have added ‘metamorphosed’

“Experimental petrology has shown that partial melts of amphibolite and eclogite are identical to TTG suites in terms of major oxides, such that TTG suites can be considered as the products of fluid-absent melting of enriched hydrated and metamorphosed basalt^{13,18–23}.”

I. 45-47 – hard to understand for none specialists. Explain that garnet is a residual phase – that is links to HREE depletion in the melt, . . . It’ s mostly reformulation. Also, provide the reader with some numbers (what kind of pressure are we talking about here?).

Thanks. We have reformulated the statement merely as the suggestion. The specified pressure was added too.

“As important peritectic minerals in the residue that control melt trace element composition (e.g., HREE, Sr, and Y), garnet (rich in HREE) is stable at a higher pressure (>1.5 GPa) than plagioclase (rich in Sr), with this variation in residual mineralogy having been used to divide TTG suites into high-, medium- and low-pressure (HP, MP, LP) types based on their La/Yb and Sr/Y ratios^{2–3}.”

Paragraph 3.

I. 51 – reformulate the beginning of this sentence

We have reformulated this sentence.

“Nonetheless, the depth of crustal melting is not the only factor controlling the Sr/Y and La/Yb compositions of the melt. Recent work suggests that continental-crust-forming magmatism during the Archaean was likely more complex than simple, single-stage fluid-absent melting^{10–11,24–35}.”

I. 51 – that’ s new, ‘continental-crust-forming magmatism’ – would be nice to introduce that earlier (that the C.C. was formed by TTG magmatism).

We have introduced this word in the first paragraph.

“Among these continental-crust-forming magmatic models, partial melting of subducted slabs or the lower portions of thickened oceanic crust are the most widely cited mechanisms for the generation of TTG magma^{4–5}.”

I. 52 - 'more complex' - why? What makes you think that? (provide rational behind every assertion)

In fact, the whole paragraph is talking about this complexity. The reason for assertion is addressed in the sentences that follow.

I. 53 - what are these drips? It's confusing, unclear what geodynamic setting we are talking about here.

We have added the description of drips.

"Fluids would have been abundant in hot subduction zone settings^{10,24-26}, and greenstone belt material may have 'dripped' into the underlying mantle to introduce considerable amounts of fluids, where such vertical motion in the Archaean Eon may differ from the dominantly horizontal motion of plate tectonics¹¹."

I. 54 - Sentence on Plag. - we have now jumped to another idea and we are lost (need to locate us: here we are talking about lower crust FP, I am guessing anyway).

No, in this sentence, we are still talking about partial melting.

I. 57 - then differentiation can modify melt X - OK, but how does it relate to the problem at hand? Will it modify the HREE content so much that we won't be able to know if garnet is a residual phase? The next sentence provides the beginning of an explanation (re-organize).

Thanks. We have added an explanatory sentence.

"Appreciable amounts of plagioclase accumulation and fractional crystallization of hornblende would induce HREE depletion and enhance Sr/Y in the melt, similar to that of HP TTG magma³²⁻³⁵."

I. 61 - if this paragraph is about the Sr/Y ratio, then say it in the first sentence (introduction sentence).

Thanks. We have now stated this at the beginning of this paragraph.

"Nonetheless, the depth of crustal melting is not the only factor controlling the Sr/Y and La/Yb compositions of the melt."

Paragraph 4.

I. 64 - before you jump to that, you need to state the problem and the aim: we need a better proxy / then you can say: we will be testing a new one (Ba). And what is petrological modelling doing here? The whole sentence is hard to follow (reformulate)

We have reformulated this sentence.

"Given such considerable variation, the Sr/Y ratio of TTG suites is increasingly being considered as an unreliable indicator of the depth of crustal melting. Instead, a pressure-dependent indicator of the original melts that is not

influenced by subsequent magmatic processes (i.e., assimilation of mantle rock and/or sediments, fractional crystallization, and accumulation) would be suitable for interpreting the tectonic setting of TTG magma formation.”

I. 66-70 - why not say that earlier, where the chemistry of TTG suites is presented? Isn't this interpreted as the involvement of a mantle component? Anyway, the whole thing should be moved up in the introduction. The whole section on MgO should have its dedicated paragraph and then the problem should be stated clearly.

Thanks. We have moved this up in the Introduction, where the chemistry of TTG suites is presented.

“TTG suites are typically rich in silica (commonly $\text{SiO}_2 > 70$ wt. %), have $\text{K}_2\text{O}/\text{Na}_2\text{O} < 0.5$, and have low concentrations of ferromagnesian elements ($[\text{MgO} + \text{FeO} + \text{TiO}_2 + \text{MnO}] < 5.0$ wt.%), with an average Mg# (= atomic $\text{Mg}/[\text{Mg} + \text{Fe}] \times 100$) of 43 (ref. ²; Fig. 1a). However, neither experimental petrology nor thermodynamic modelling has been able to satisfactorily simulate the production of such high-MgO TTG suites solely by the partial melting of a basaltic protolith¹³⁻¹⁵. Some MgO-rich melts or rocks (e.g., komatiite, picrite, and peridotite) may need to be assimilated into a TTG magma or into the source rock to produce the observed compositions¹⁶⁻¹⁷.”

I. 70-71 - not clear. Modelling cannot generate melt with enough MgO, so fractionation must be limited? What is the link, what is fractionation doing here if we cannot even reproduce the primitive melt (such experiments tell us nothing on the extent of fractionation).

Thanks. We have deleted this sentence.

I. 73 - yes, mantle involvement, you are not the first to suggest that (see work of Martin for example).

Thanks. Martin & Moyen's paper (2002, Geology) was cited.

Martin, H., & Moyen, J. F. Secular changes in tonalite–trondhjemite–granodiorite composition as markers of the progressive cooling of Earth. Geology 30, 319–322(2002).

I. 76 - what mass balance calculation? Need reference and explanation. Well. In fact, the mass balance calculation is done by us. It is easy to do this. The Ba content of MgO-rich rocks is lower than that in primitive melt. When MgO-rich rock mixes into the primitive melt, the Ba content of the mixed magma could not be higher than the primitive melt. Therefore, mixing can only decrease the Ba content of magma.

I. 77 - where does this come from?

We have reorganized this paragraph, and broken it into two. We want to demonstrate that the characteristic of high Ba contents in TTG is primitive and could not be enhanced by later magmatic progresses (e.g., assimilation of mantle rock or sediments and fractional crystallization or accumulation). As such, we will rule out these possibilities one by one.

“Given such considerable variation, the Sr/Y ratio of TTG suites is increasingly being considered as an unreliable indicator of the depth of crustal melting. Instead, a pressure-dependent indicator of the original melts that is not influenced by subsequent magmatic processes (i.e., assimilation of mantle rock and/or sediments, fractional crystallization, and accumulation) would be suitable for interpreting the tectonic setting of TTG magma formation. As an incompatible element, Ba has a rather low concentration in magnesium-rich rocks that are often quite primitive. As described, based on mass balance calculations, the assimilation of MgO-rich rocks into a magmatic system would not cause an increase in Ba content of the mixed magma. It then follows that the assimilation of sediments, rich in large ion lithophile elements (LILE; i.e., K, Rb, Sr, Ba, etc.), could enhance the Ba content in TTG magma³⁶. If the assimilation of sediments were to explain elevated Ba enrichment in TTG magma, then their LILE and Ba contents should be coupled. However, most high-Ba TTG suites are characterized by low K₂O/Na₂O (Fig. 1b), as well as low Rb. This compositional decoupling thus excludes the possibility of sediment assimilation to account for Ba enrichment. Even though plagioclase accumulation would efficiently change the Sr/Y ratio in TTG magma, it cannot enhance the Ba content of a melt, as the partition coefficient between plagioclase and granitic melt is very close to 1 (refs. ^{4,35}). Of all rock-forming minerals likely involved in Archaean crust formation processes, only biotite exhibits strong compatibility with Ba; however, biotite accumulation during TTG melt production, ascent, and crystallization is also unlikely. Importantly, Ni is also compatible in biotite, but Ni contents in most high-Ba TTG suites (Ba > 1,000 ppm) does not exceed 30 ppm, which is even lower than those of low-Ba TTG suites (Fig. 1d). Therefore, crystal accumulation cannot explain these compositional features, and the only remaining possibility would be a diversity of original melt compositions.

In this work, we present and test a new compositional proxy—barium (Ba) contents in TTG suites—for interpreting the tectonic setting of TTG magma formation by conducting petrological modelling of partial melting of a basaltic composition²⁶. It has been discussed that TTG suites are highly enriched in large-ion lithophile elements (LILEs; e.g., K, Rb, and Ba), and cannot be generated from mid-ocean ridge basalts (MORB) and oceanic plateau basalts (OPB), which are depleted or not enriched enough in LILEs^{26,37}. Oceanic island basalts (OIB), even though they are enriched in LILEs²⁶, also cannot generate large volumes of TTG magma owing to their restricted abundance. Ge et al. ²⁶ compiled more than 1,000 enriched Archaean tholeiitic basalts that

are characterized by moderately enriched LILE and have Th/Nb > 0.1 and suggested these basalts would be a suitable source for generating large volumes of TTG magmas. These basalts have a wide range of trace element compositions, and would not necessarily be related to subduction since the oblique array joining a rather depleted portion of the mantle array to the arc field would result from continental crust assimilation³⁸. To verify the hypothesis that the diversity of original melt compositions accounts for high-Ba TTG magmas, the average enriched Archaean tholeiitic basalt composition²⁶ was used to carry out petrological modelling at different geothermal gradients and water contents, which allowed predictive modelling of melt compositions in different geodynamic settings. The average documented Ba content of enriched Archaean tholeiitic basalts is 107 ppm, and most have less than 160 ppm, indicating that there is not enough variability in the primary Ba contents of Archaean basalts to account for the diversity of Ba contents in TTG suites, especially in the early Archaean when general compositional diversity (e.g., in Ba) is predicted to have been lower²⁶.”

l. 78 - what are sediments doing here now? / l. 82 - and there goes sediments. It is very confusing - what model are we testing here? Subduction? Then start from subduction, from what we can expect in such setting, and discuss TTG chemistry by always stating what it means for the subduction model.

As we said above, assimilation of sediments is a possibility that would influence the Ba content in magma. We have reorganized this paragraph. We hope the revised version is more logical.

Idem for the rest of the paragraph, need reorganizing. And all remarks need to be explained and referenced (e.g., l. 86 - Biotite accumulation is unlikely - based on what evidence?).

Thanks. We have reorganized this paragraph. Necessary explanations and citations were added. See revision two comments above.

This section is very confusing because it concludes that Ba is the best proxy: then work done, let's conclude the paper. You need to reorganize all this and to orient the paper toward the conclusion that is stated here: Ba is best because, according to our Results, it is a proxy to the composition of the primitive melt (that's for the Discussion section, not the Introduction section).

We are sympathetic to the logic suggested by reviewer #3, however, we must explain why we follow the logical structure we do. We first need to know why we should test Ba instead of other elements. Ba could reflect the primitive composition of the melt. Then, we could use Ba content evolution of the melt to identify the tectonic setting of TTG formation. Our modeling results are for primitive compositions. Therefore, we need to evaluate the importance of Ba at the beginning. Hopefully our rationale in the considerably reorganized final

paragraph of the Introduction (with its criteria clearly stated in the formatting of Nature Communications papers) is now more clear to the reviewer.

L. 90 – referring to the EAT may be done later – you need to state where this composition comes from (reference).

Done. The EAT composition was cited.

“In this work, we present and test a new compositional proxy—barium (Ba) contents in TTG suites—for interpreting the tectonic setting of TTG magma formation by conducting petrological modelling of partial melting of a basaltic composition²⁶. It has been discussed that TTG suites are highly enriched in large-ion lithophile elements (LILEs; e.g., K, Rb, and Ba), and cannot be generated from mid-ocean ridge basalts (MORB) and oceanic plateau basalts (OPB), which are depleted or not enriched enough in LILEs^{26,37}. Oceanic island basalts (OIB), even though they are enriched in LILEs²⁶, also cannot generate large volumes of TTG magma owing to their restricted abundance. Ge et al.²⁶ compiled more than 1,000 enriched Archaean tholeiitic basalts that are characterized by moderately enriched LILE and have Th/Nb > 0.1 and suggested these basalts would be a suitable source for generating large volumes of TTG magmas. These basalts have a wide range of trace element compositions, and would not necessarily be related to subduction since the oblique array joining a rather depleted portion of the mantle array to the arc field would result from continental crust assimilation³⁸. To verify the hypothesis that the diversity of original melt compositions accounts for high-Ba TTG magmas, the average enriched Archaean tholeiitic basalt composition²⁶ was used to carry out petrological modelling at different geothermal gradients and water contents, which allowed predictive modelling of melt compositions in different geodynamic settings. The average documented Ba content of enriched Archaean tholeiitic basalts is 107 ppm, and most have less than 160 ppm, indicating that there is not enough variability in the primary Ba contents of Archaean basalts to account for the diversity of Ba contents in TTG suites, especially in the early Archaean when general compositional diversity (e.g., in Ba) is predicted to have been lower²⁶.”

Paragraph 5.

Line 96. Ba content of what? I don't expect the whole rock Ba composition to change much during metamorphism.

Ba content of melts. Sorry for this misleading statement, which has been clarified.

“Given that hot subduction and lower crustal drips are the most accepted geodynamic settings for Archaean TTG magma generation, two typical geothermal gradients of 450 GPa⁻¹ and 900 °C GPa⁻¹, respectively, are considered for investigating the evolution of the Ba content of the melts during metamorphism^{2, 39–41}.”

I. 100. Would be nice to start with that (what are we trying to model?). Then move to how the model is performed, and describe the results.

Thanks. As suggested by reviewer #3, we have reformulated this paragraph. The possible environments for TTG generation were introduced at the beginning of the paragraph.

“Given that hot subduction and lower crustal drips are the most accepted geodynamic settings for Archaean TTG magma generation, two typical geothermal gradients of 450 GPa⁻¹ and 900 °C GPa⁻¹, respectively, are considered for investigating the evolution of the Ba content of the melts during metamorphism^{2, 39–41}.”

I. 104. ‘hot subduction’ – might be best to refer to model A, B, . . . - or start the paragraph like this: model A has these characteristics and aims at modelling the hot subduction scenario, . . . (reorganizing).

Thanks. We have reorganized this paragraph, just as suggested by reviewer #3.

“Given that hot subduction and lower crustal drips are the most accepted geodynamic settings for Archaean TTG magma generation, two typical geothermal gradients of 450 GPa⁻¹ and 900 °C GPa⁻¹, respectively, are considered for investigating the evolution of the Ba content of the melts during metamorphism^{2, 39–41}. As an additional possibility, a geothermal gradient of 250 °C GPa⁻¹, corresponding to a cold subduction setting^{39,42}, was also explored. According to the assumption that fluid-absent melting is the main mechanism to generate TTG-like melts¹⁵, previous work has explored the evolution of melt composition under water-unsaturated conditions. To investigate the evolution of Ba content in melts under water-saturated conditions, based on the possibility of fluid-present melting described earlier and to make a comparison with melts under water-unsaturated conditions, a *P–T* diagram for an enriched Archaean tholeiitic basalt composition was produced with an H₂O content of 7.0 wt. % to ensure all the calculated melts were generated through fluid-present melting (Fig. 2a). Compared with the *P–T* diagrams calculated at water-reduced conditions¹⁵, the rocks with higher water contents are much more fertile (Fig. 2, b and c). For the hot subduction setting, approximately 50 vol. % of melt is predicted to be generated per unit volume of protolith at hot subduction zones if sufficient water is present, compared to ~20 vol. % in a water-limited environment. Enhanced water contents also suppress the stability field of hornblende and quartz to lower temperatures, and these minerals are preferentially and efficiently consumed at relatively low temperatures (Fig. 2b). For an Archaean drip setting, plagioclase stability would be suppressed to lower pressure and quartz to lower temperature (Fig. 2c). While amphibole tends to remain stable before quartz and plagioclase are consumed. These mineralogical differences in residue lead to a diverse range of melt composition evolutions. For a cold

subduction setting, calculated melt proportion isopleths lie almost parallel to the geothermal gradient, leading to no more than 4 vol.% of melt generation (Supplementary Fig. 6), which would also not be able to be extracted from the protolith⁴³. Such results can thus rule out cold subduction for TTG magma generation.”

l. 110 - . . . according to tectonic setting. You should separate things: first talk about modelling and its results, and later in the text relate results to geological realities. Thanks. This paragraph was reorganized, and can be seen immediately above.

Paragraph 6.

l. 114 – I am confused, I thought only one protolith composition (EAT) was tested? Do you mean water-content?

Yes, the differences derive from the water content.

“Predicted melt compositions are thus a function of the P – T conditions of generation, as well as their protolith compositions, including water content.”

Paragraph 7.

The first sentence has nothing to do here – it doesn’t do a good job introducing a paragraph that describes primitive melt composition. Focus this paragraph on describing Results.

The first sentence was deleted.

l. 136-etc. – need to cover that earlier in the text. A section dedicated to presenting the EAT source and discussing Ba distribution in Archean basalts and TTG would be useful – to present before presenting the Results of the model. We have moved this statement before the Results (at the end of the final paragraph of the Introduction summarizing the results of the study).

“The average documented Ba content of enriched Archean tholeiitic basalts is 107 ppm, and most have less than 160 ppm, indicating that there is not enough variability in the primary Ba contents of Archean basalts to account for the diversity of Ba contents in TTG suites, especially in the early Archean when general compositional diversity (e.g., in Ba) is predicted to have been lower²⁶.”

Paragraph 8.

This is not located in the right place. You need to dedicate part of your text (and only one part of it) to the different possible geodynamic scenarios. Idem for line 157 – that is the fourth time we get back on this fractionation issue – you need to discuss fractionation once and for all. The whole text needs reorganizing. After reorganizing the text, this paragraph now appears in the beginning of the Discussion, as suggested. For the idem for line 157, we have deleted the mention of fractionation here.

“It is widely accepted that the mantle potential temperature in the early Archaean was higher (~250 °C) than that at present^{44–45}. Secular cooling of the ambient mantle would influence coupling across the lithosphere-asthenosphere boundary, possibly leading to a transition in the tectonic regime from a stagnant lid to modern plate tectonics⁴⁶. Even so, the age of this transition is debated, with propositions ranging from the Eoarchaean to Neoproterozoic^{26,47–49}. Stagnant-lid regimes differ from mobile lid (plate tectonic) regimes by being dominated by vertical tectonic motion and deformation (e.g., dome and keel; sagduction; drip), as opposed to horizontal tectonic features (e.g., linear collisional orogens). Well-preserved vertical structures could be recognized at some older cratons, such as eastern Pilbara cratons in western Australia, Kaapvaal and Zimbabwe cratons in southern Africa^{50–52}, although they also coexist with linear collisional orogens at some younger cratons, like the Superior craton, Canada and North China craton^{53–54}. Geodynamic modelling shows that the P – T paths of rocks within these vertical structures follow a relatively high geothermal gradient⁴¹. It is therefore plausible that only subduction zones would provide a sufficiently low geothermal gradient to produce Ba-rich magmas. Ba contents in TTG suites would not be influenced during fractional crystallization and magma assimilation, and the diversity of Ba contents in the source (enriched Archaean tholeiitic basalt) would be limited as described. Thus, it would be a sensitive indicator of the tectonic setting of formation.”

Paragraph 9.

I. 162. Not clear that Ba is now establish as a proxy for subduction (not tectonic setting according to what I read before).

We got this from the Results and the first section of Discussion.

“It is therefore plausible that only subduction zones would provide a sufficiently low geothermal gradient to produce Ba-rich magmas. Ba contents in TTG suites would not be influenced during fractional crystallization and magma assimilation, and the diversity of Ba contents in the source (enriched Archaean tholeiitic basalt) would be limited as described. Thus, it would be a sensitive indicator of the tectonic setting of formation.”

See remark made earlier. It would be more pertinent to first present what Ba looks like in these rocks, and to then move on to modelling, rather than doing it the other way round.

This comment is addressed above in response to the earlier comment about organization.

Would be nice to say a word on the dataset – are we looking at a global compilation? Done by who?

Yes, this is a global compilation, done by Johnson et al. (2019). We have added the citation.

“Having established Ba contents as a proxy for the tectonic setting of TTG magmatism, we test for the presence of any potential shifts in the global record of Archaean TTG suites⁶ that might indicate the evolution of geodynamic regimes.”

Paragraph 10.

I. 170. ‘indicates their formation in an intraplate setting’ – according to your model. This is a major flaw of this paper: too many affirmations that are lacking context or references. And what does ‘intraplate setting’ look like for Archaean TTG?

Sorry. We have changed the intraplate setting to an Archaean drip setting.

“Before 3.7 Ga, with the sole exception of Slave craton, nearly all TTG suites preserved in the rock record have low Ba contents, which indicates their formation in an Archaean drip setting.”

I. 173. ‘become complicated’ – reformulate / ‘change point’ or ‘turning point’ (there is a GSA volume that should come out soon that discusses the ‘turning point’ concept for the North American craton).

We have changed the phrase ‘change-point’ to ‘positive shift’

“The Ba contents in TTG suites show a statistically significant positive shift at 3.7 Ga and do not show an additional step change thereafter, implying an irreversible change took place in global geodynamics (Fig. 4a).”

I. 175 – why ‘delayed’? They are different (diachronicity)

We have changed the word ‘delayed’.

“The apparent “global” 3.7 Ga positive shift of all TTG suites is in verity likely a signature of the North Atlantic craton, while the ages of shifts in other cratons are typically younger and highly diachronous (Fig. 4a).”

I. 176. The TTG suites of the Slave craton (not the Slave craton)

We have reformulated this sentence.

“The TTG suites of the Slave craton preserve high-Ba values as early as 4.0 Ga, which it maintains throughout the rest of the Archaean.”

I. 178-179 – reformulate (and don’t repeat ‘Slave craton’ so many times) We have reformulated this sentence.

“A strongly positive Ba anomaly has been identified at 3.5 Ga in Slave (Fig. 4a), which, if interpreted as an indicator of subduction, is independently corroborated by a seismically imaged slab beneath the Slave craton dated to ca. 3.5 Ga (ref. ⁶¹).”

I. 180 – useless sentence (it is what you suggest in the whole section) We have deleted this sentence.

I. 181. Remove ‘been’

We have deleted the word.

“Subduction might have occurred in the Slave craton since 4.0 Ga, followed by initiation at 3.7 Ga in the North Atlantic craton, while most others initiated at 3.2–3.0 Ga (e.g., Baltica, Amazonia, and Pilbara), or even much later at 2.7 Ga (Superior craton).”

I. 184. We are going back on this fractionation things (fifth time) – reorganize

We have move this sentence to where fractionation first appears.

“Appreciable amounts of plagioclase accumulation and fractional crystallization of hornblende would induce HREE depletion and enhance Sr/Y in the melt, similar to that of HP TTG magma^{32–35}. ”

I. 188. Stagnant-lid vs plate tectonic are notions to be introduced much earlier in the text

Stagnant-lid vs plate tectonics is introduced at the beginning of the Discussion.

“Stagnant-lid regimes differ from mobile lid (plate tectonic) regimes by being dominated by vertical tectonic motion and deformation (e.g., dome and keel; sagduction; drip), as opposed to horizontal tectonic features (e.g., linear collisional orogens).”

I. 190-194. Here you are saying that other data (like mantle cooling) are useless so we need a new proxy – this should be said in the Introduction. Here, most of the paragraph talks about diachronicity – mantle cooling is global and cannot explain diachronic event. So what can explain this diachronism?

Although we appreciate the reviewer’s recommendation about paper structure, we nonetheless view mantle cooling as more of an implication (apt for Discussion) than a precept of this study (for an Introduction).

We very much appreciate the reviewer asking us to expand upon our explanation for the observed diachroneity. There is no reason mantle cooling should be global unless whole mantle convection can be asserted—which is unproven, and would be entirely theoretical, at this old age. It is therefore not only possible, but also likely, that localized mantle cooling—where convective sinking occurring in much smaller regions in the Archean as compared to present-day Earth’s broad regions like the Pacific ring of fire—could account for the observed diachroneity. More detailed study of cratonization patterns should be done in the future to test this idea. We have expounded on this explanation in the revised text.

“The initiation of the subduction would be a function of several factors, including mantle temperature, radiogenic heat production, density contrast between the lithospheric mantle and the convective mantle, and lithospheric thickness⁶². The whole Earth would not be homogenous in all these factors,

so we may not expect the onset of subduction to be a globally isochronous event as has often been portrayed in past work. Subduction likely is initiated once all these factors are suitable, which should naturally only be regional in scale at the beginning, and eventually evolve to a global scale once other regions are suitable⁶³.”

I. 195-197. These 2 sentences make no sense: if the oldest rocks record the shift, then there are no older rocks to record low Ba content. And what is ‘rock preservation’ doing here? If there is no older TTG suite, could be they never existed (doesn’t necessarily mean they got eroded).

We apologize for the misunderstanding and have revised the text to make it clear that two different ages are being correlated for each craton: the age of their oldest rocks and the age of their Ba shift. (So it is not saying “the oldest rocks record the shift” as the reviewer understood.) We hope the revised text makes more clear. We have also revised what this fascinating correlation—those cratons with older rocks have earlier Ba shifts—implies for the potential relationship between rock preservation and subduction.

“Furthermore, the ages of Ba positive shifts of various cratons correlate strongly with the ages of the oldest rocks of each craton (Fig. 4b). According to our proposal that Ba positive shifts indicate subduction initiation, this correlation would thus imply an important relationship between rock preservation and subsequent subduction.”

Paragraphs 10 + 11 + 12

I don’t see the plus-value of these data, especially at this stage of the demonstration. These are statements for the Introduction. Also, given that Palin et al. (2020) recently reviewed many such evidences, I would rather shorten and move this paragraph to the Introduction – alternatively, just delete it.

Thanks for this comment, we have simplified these paragraphs into a short paragraph at the second paragraph of the Discussion section.

“Whenever a new compositional proxy for subduction is introduced, its broader implications must be considered within a multi-proxy context. Based on different proxies, however, the timing of the onset of plate tectonics is contentious^{47–49,55–57}. Most of the proxies either record the first appearance of specific rock types or metamorphic facies (e.g. blueschist, Precambrian paired metamorphism)^{49,58–59}, or indirectly infer the nature of continental crust^{47–48}. Such data are usually considered as the upper limit age of onset plate tectonics. Archaean basalts and komatiites might suggest a re-enrichment of Earth’s mantle at 3.2 Ga⁵⁵, but dating such silicon undersaturated rocks is difficult. In contrast, the oldest granitoids on Earth can be precisely dated and formed at ca. 4.0 Ga in Slave craton (Acasta TTG suites⁶⁰). Eoarchaean to Palaeoarchaean (4–3.2 Ga) TTG suites also occur in many cratons worldwide (e.g., Greenland, Barberton, and Pilbara), providing more information on the earliest Earth. Ba content in TTG suites shows more affinity for initial partial

melting than later magmatic effects, strongly indicating that Ba concentration is a more reliable indicator than other element contents or ratios of the primary melt and associated tectonic setting. A multi-proxy understanding of subduction initiation on Earth may remain elusive, but the results presented here for Ba provide a path forward with a more unequivocal proxy. We suggest such a new proxy can provide the framework for developing a multi-proxy approach in the future.”

Idem for paragraphs 11 and 12 - this sounds like an Introduction.

Thanks. We have simplified these paragraphs into a short one, and could be seen at the second paragraph of the Discussion. We think we should compared the pros and cons between different proxies, which could show the advantage of Ba content in TTG suite. Therefore, we'd like to leave it in the Discussion.

Figure 4. Figure 4a is very important to the text, but it is unreadable - it is hard to see what you mean by change-point and which data correspond to which craton. Need to revise the design of this Figure.

Thanks. Figure 4a have been replaced, and could be seen below.

Figure 5. Useless Figure (this has been published)

To our knowledge, this version of the mantle cooling curves depicting different ages for the onset of plate tectonic convective cooling has not been published anywhere and is therefore retained—particularly as our constraints from Ba allow these various models to be weighted for their relative success in satisfying this new constraint.

REVIEWER COMMENTS

Reviewer #1 (Remarks to the Author):

Second review of manuscript by Guangyu Huang et al,

This is a significantly revised and much improved manuscript that has certainly gone a long way to satisfying or clarifying most of the comments posed by the reviewers. It flows well and is now a pleasure to read. The idea and arguments put forward supporting the use of Ba in TTG as a proxy for a globally diachronous change from vertically-dominated to horizontally dominated process are very interesting and potentially a significant step forward. I remain slightly sceptical that this proxy is as robust as stated, but the idea certainly needs to be presented.

I have again made several comments and suggestions on the edited manuscript – but most at the editors/authors discretion, and am otherwise satisfied that publication can proceed.

Reviewer #3 (Remarks to the Author):

This manuscript reads well and the topic discussed is of interest to a broad audience. Please find, below, some comments for your consideration.

Kind regards.

l. 24. 'difficult to enhance after magma generation' – don't know what you mean, please

clarify l. 26 'thus representing' – reformulate (thus Ba represents...)

l. 51-53 – this makes no sense at all – if komatiite were assimilated by TTG melt, the median SiO₂ value wouldn't be 70 wt%.

l. 62-65. Al and Na content have been used too to comment on partial melting depth. l. 82 – on the differentiation of TTG suites, see also Mathieu (2022)

l. 115... - MORB, OIB, ... categories may not apply to Archean melts (see the work on J-F Moyen and O. Laurent on this matter)

l. 129 – these are average values, we have no information on data spread... - so you can't argue that it wasn't heterogeneous enough to account for Ba variation in TTG suites. The basalt composition dataset (through time) should be presented in more details.

I. 201 – It would be nice to model fractional crystallisation (to demonstrate that it really cannot impact the Ba content of TTG suites)

I. 253-255 – see also Mathieu et al (2022) – GSA special volume on Turning Points

I. 257 – on diachronicity, would be nice to discuss this by referring to Laurent et al. (2014) – are the timing of your shifts matches this of Laurent’s shifts?

Reviewer #4 (Remarks to the Author):

The paper by Huang et al. presents a novel concept of using Ba concentrations of Archean silicic crustal rocks as a proxy for the geodynamic setting of magma generation. They use petrological modelling to predict Ba contents of melts produced through Archean basalt melting under different geothermal gradients, and compare the results with global geochemical data on the most common Archean granitoids (TTGs) to highlight systematic variations in Ba content in these rocks. They argue that TTGs from all cratons worldwide show a shift of increasing Ba contents with time, which they interpret as reflecting the onset of subduction; and further discuss that this shift is diachronous from craton to craton, highlighting how plate tectonics started to operate progressively on a global scale.

The topic is high-profile and suited to publication in Nature Comm. The idea of using Ba as a proxy is smart, as it is a high-concentration, widely available element even from relatively old datasets. The petrological model is well designed and the results are realistic. Altogether, the paper portrays a very reasonable and sensible view of the global geodynamic evolution throughout the Archean, consistent with that obtained using other constraints. Although the paper went already through a round of review and the authors did satisfactorily address the comments, I would have further suggestions for moderate revision and improvement before it can be accepted:

1. The strongest assumption underlying the authors’ reasoning is that Ba is not affected by any possible compositional evolution of the magma after formation in the source (e.g. fractional crystallization, assimilation etc.). In turn, the authors assume that there are Ba-rich vs. Ba-poor TTG suites. In fact, one could also argue that Ba-poor vs Ba-rich TTGs represent different end-members of single TTG suites linked by differentiation processes. I don’t think this is necessarily the case, but to validate the model, the authors should challenge their hypotheses against the natural TTG record. For instance, they could use the data of our recent paper on the Paleoproterozoic Barberton TTGs (Laurent et al., 2020) which, I think, strengthen their point as the least chemically evolved tonalites-diorites have similar Ba compositional range as the most evolved granitic end-member of the suite. There might be more examples to investigate in the literature (in fact, the TTG dataset from Moyen 2011 is somewhat outdated, as a lot of work has been done to understand the chemical diversity of TTGs worldwide over the past 10 years) and I encourage the authors to elaborate on this part of the discussion.

2. In relation to the previous point: the major element composition of the melts produced in the models are hardly compared with those of natural TTGs – just adding natural TTG compositions to Fig 3 would already help a lot.

3a. About the statistical evaluation of the global TTG dataset: in terms of Ba concentrations, the real distinctive difference between the modelled low vs high geotherms is that high-Ba TTG-like melts (>800 ppm) are restricted to the low geotherms. Anything with lower Ba is basically non-discriminant. I agree with the authors that the ‘upward steps’ in the Ba time series must indicate that such high-Ba TTGs got more common in the record with time. However, there is considerable scatter in the Ba content of TTGs at a given time (often an order of magnitude or more), so it is therefore difficult to evaluate whether the steps result from the occurrence of only very few outliers at very high Ba, or a more generalized yet modest increase of Ba contents. I would love to see a more statistically rigorous way of representing the Ba range at a given time, for instance with boxes-and-whiskers indicating the interquartile range + outliers. Perhaps using median values, instead of averages, would be also more robust as the data follow a non-normal distribution. I’m pretty convinced that this would not change significantly the results, but IMHO this would make the argumentation more compelling.

3b. In complement to this, from Supp Figure 5 it is like the average Ba content of some time slices is based on very few points (perhaps only one sometimes?) while others are much better constrained. Does the time-resolved Bayesian analysis include some sort of weighing to account for this? If yes, this should be explained in the methods section, and if not, it should be mentioned too. In the latter case, it might be worth running also the algorithm without the under-represented time slices and check whether the results are significantly different, or not.

4. Last but not least, the authors consider Ba contents as “a new compositional proxy for subduction” (l. 205). In fact, based on their model, Ba contents should be strictly regarded as a proxy for “melt production under a low geothermal gradient” – whether this resulted from subduction as we know it from modern plate tectonics, or some form of more primitive, local and/or transient process of crust burial at mantle depths (see models & papers by Taras Gerya et al., Jean Bédard et al., and discussion in Moyen & Laurent 2018) is actually a different, and unaddressed question. I guess discussing this in detail is beyond the scope of this paper but the nuance should be clearly expressed and perhaps the claim of a “proxy for subduction” somewhat tuned down.

Besides these main points, I listed a few additional, minor comments in the attached commented version of the paper. I am looking forward to see this work published.

Sincerely,

Oscar LAURENT – Toulouse, 07.07.2022

Responses in blue

Revisions in red

REVIEWER COMMENTS

Reviewer #1 (Remarks to the Author):

Second review of manuscript by Guangyu Huang et al,
This is a significantly revised and much improved manuscript that has certainly gone a long way to satisfying or clarifying most of the comments posed by the reviewers. It flows well and is now a pleasure to read. The idea and arguments put forward supporting the use of Ba in TTG as a proxy for a globally diachronous change from vertically-dominated to horizontally dominated process are very interesting and potentially a significant step forward. I remain slightly sceptical that this proxy is as robust as stated, but the idea certainly needs to be presented.
I have again made several comments and suggestions on the edited manuscript – but most at the editors/authors discretion, and am otherwise satisfied that publication can proceed.

Reviewer 1 made some correction on the manuscript. We all accepted. Here are responses to the comment on edited manuscript.

Line 35: remove ‘essentially identical’

Revision: Thanks. We have removed it.

Line 40: oceanic in what sense? Simply because its under water? Why not just ‘mafic crust’?

Revision: We have change it to ‘mafic crust’.

Line 92: change ‘magnesium’ to ‘MgO’

Revision: We have changed it.

Line 92: change ‘As described, based on...’ to ‘Based on...’

Revision: We have changed it.

Line 94: change ‘It then follows that...’ to ‘However, ...’

Revision: We have changed it.

Line 114: ‘LILEs’ Already introduced (line 95)

Revision: We have changed it.

Line 123-124: I presume you are talking about the Th/Yb-Nb/Yb Pearce plot – say so and rephrase this sentence

Revision: Yes, the reviewer's presumption is correct. We have rephrased for clarity:

"These basalts have a wide range of trace element compositions, and would not necessarily be related to subduction since the oblique array in the Th/Yb-Nb/Yb plot joining a rather depleted portion of the mantle array to the arc field would result from continental crust assimilation³⁹."

Line 129-132: Yes – but you are assuming 'enriched' basalts formed the entire melting pile. Many Archaean mafic sequences are VERY inhomogeneous – so why not various combinations of enriched and unenriched basalts adding to original diversity. Just a comment.

Revision: Thanks. We have added some details of this data:

"The average documented Ba content of enriched Archaean tholeiitic basalts is 107 ppm, with a median value of 62 ppm. Among the more than 1,000 Ba content data, nearly 90% of these basalts have less than 160 ppm, indicating that there is not enough variability in the primary Ba contents of Archaean basalts to account for the diversity of Ba contents in TTG suites, especially in the early Archaean when general compositional diversity (e.g., in Ba) is predicted to have been lower²⁶."

Line 137-139: Agreed – dealing with these two setting for now is good enough.

Response: We are glad the reviewer agrees with our chosen scope.

Line 150-151: Is there some way you can actually make this comparison on these diagrams? Rather than requiring readers to go to ref. 15

Response: Readers can compare $P-T$ diagrams in Supplementary Figures 1-4.

Revision: We have added this description and citation to the relevant figures: "Compared with the $P-T$ diagrams calculated at water-reduced conditions¹⁵ (Supplementary Fig. 1-4), the rocks with higher water contents are much more fertile (Fig. 2, b and c)."

Line 153: Remove 'at hot subduction zones', not needed

Revision: We have removed it.

Line 196: change 'could be' to 'can be', 'are'

Revision: We have changed it.

Line 198: What do you mean by 'coexist'? are you saying that BOTH occur in the Superior and NCC?

Revision: Yes. We have clarified here and also made some edits based on the comment from reviewer #4:

"Well-preserved vertical structures can be recognized at some older cratons, such as eastern Pilbara cratons in western Australia, Kaapvaal and Zimbabwe cratons in southern Africa⁵¹⁻⁵³, although individual cratons do show a succession

of ‘dominantly vertical’ followed by ‘dominantly horizontal’ regimes throughout their evolution.”

Line 201-202: On its own, this statement reads like a leap of faith. What about ‘Because our data suggests, it seems plausible that..’”

Revision: Thanks. We have changed it as suggested.

Line 214: Archaean basalts and komatiites don’t suggest anything – but ‘Compositional changes in Archaean basalts.....’ might!!

Revision: Thanks. We have changed it as suggested.

Line 215: Directly, yes it is! But lets not sensationalize this and upset a large group of researchers who clearly have obtained quite good constraints on most greenstones throughout nearly all Archean terranes!!!! – just a comment but I suggest you modify this statement.

Revision: Thanks. We have rephrased this sentence:

“Compositional changes in Archaean basalts and komatiites might suggest a re-enrichment of Earth’s mantle at 3.2 Ga⁵⁴, but dating such silica undersaturated rocks is not as easy as felsic rocks.”

Line 221-222: This simply repeats the first part of the sentence

Revision: We have removed it.

Line 222: Why are “later magmatic effects” (other than post crystallization alteration) any less a reflection of tectonic setting?

Revision: We have removed it.

Line 229: change ‘established’ to ‘suggested’

Revision: We have changed it.

Line 234: Is it worth having tie-lines connecting the inflections in the lines for each craton to the relevant points causing them?

Response: We appreciate the reviewer’s suggestion. However, practically speaking, none of our attempts to revise, in our opinion, improved the figure significantly and introduced some unnecessary complexities. Tie lines makes the figure a bit messy, and to be understood, require color-coding the data by craton, which distracts, in your opinion, from the main results. Furthermore, readers wishing for more details on the inflections by craton can all the relevant data points causing them in Supplementary Figure 8.

Line 244: ??? meaning of ‘is in verity’?

Revision: We have reworded for clarity.

Line 247-248: So why is there no a red line traversing the ‘rest of the Archean’ rather than a dot at 4Ga?

Revision: Excellent suggestion. The red line traversing the rest of the Archean according to the data analysis of Slave craton has been added to the revised figure.

Line 263: change ‘suitable ’ to ‘satisfied’

Revision: We have changed it.

Reviewer #3 (Remarks to the Author):

This manuscript reads well and the topic discussed is of interest to a broad audience. Please find, below, some comments for your consideration.

Kind regards.

I. 24. ‘difficult to enhance after magma generation’ – don’t know what you mean, please clarify

Response: Thanks. Reviewer #4 also pointed out this. Sorry for this. We mean that the Ba concentrations would be difficult to enhance after melt generation in the source (i.e., with fractional crystallization, assimilation etc.).

Revision: We have rephrased this sentence:

“We conduct petrological modelling over a range of pressure–temperature conditions relevant to the Archaean geothermal gradient using an average enriched Archaean basaltic source composition to predict Ba concentrations in TTG suites, which is difficult to enhance after magma generated in the source.”

I. 26 ‘thus representing’ – reformulate (thus Ba

represents. . .) **Revision:** We have edited as suggested.

I. 51-53 — this makes no sense at all — if komatiite were assimilated by TTG melt, the median SiO₂ value wouldn't be 70 wt%.

Response: In fact, we cited this scenario from the reference of Martin and Moyen (2002). The SiO₂ contents of TTG are commonly >70 wt.% according to the description of Moyen and Martin (2012). But in the datasets of Moyen (2011) and Johnson et al. (2019), large numbers of TTG (nearly half) have SiO₂ <70 wt.%, some even <60 wt.%. The SiO₂ contents of TTG show a negative relationship with those of MgO and Mg# (see figure below). Most high-MgO TTG have SiO₂ <65 wt.%. It is therefore reasonable that these TTG might have assimilated high-MgO rocks like komatiite. Otherwise, based on our calculation, the SiO₂ would be always >70 wt.%.

I. 62-65. Al and Na content have been used too to comment on partial melting depth.

Response: It is true that these elements have been proposed to be able to be used in such a capacity (e.g., the classification in Moyen, 2011). Even so, we have modeled the major compositions of the melt generated at different depth. From Supplementary Table 2, the Al and Na of the melts generated at different depths from the same source composition are in the same range, and are difficult to be distinguished. In such a case, Al and Na would not be sensitive to the pressure/depth during partial melting.

I. 82 — on the differentiation of TTG suites, see also Mathieu (2022)

Revision: Good recommendation. We have cited this reference.

I. 115. . . - MORB, OIB, . . . categories may not apply to Archean melts (see the work on J-F Moyen and O. Laurent on this matter)

Response: True. We describe this possibility in our responses to comments from other reviewers, who suggested we add more details of the source composition. The source composition comes from Ge et al. (2019), and these sentences are how they filtered the data.

I. 129 — these are average values, we have no information on data spread. . . .
- so you can't argue that it wasn't heterogeneous enough to account for Ba variation in TTG suites. The basalt composition dataset (through time) should be presented in more details.

Revision: We agree. Although this paper (Ge et al., 2019) didn't publish the dataset, they provide some details of it. We would like to present as much as we can and thus expanded the relevant text accordingly:

"The average documented Ba content of enriched Archaean tholeiitic basalts is 107 ppm, with a median value of 62 ppm. Among the more than 1,000 Ba content data, nearly 90% of these basalts have less than 160 ppm, indicating that there is not enough variability in the primary Ba contents of Archaean basalts to account for the diversity of Ba contents in TTG suites, especially in the early Archaean when general compositional diversity (e.g., in Ba) is predicted to have been lower²⁶."

I. 201 — It would be nice to model fractional crystallisation (to demonstrate that it really cannot impact the Ba content of TTG suites)

Response: Good idea. The following figure presents the results of different degrees of plagioclase accumulation. The proto melt composition comes from 7 mol.% fluid-present melting at 900 °C GPa⁻¹ in this study. The plagioclase composition is calculated in equilibrium with the melt, based on the partition coefficient list in Supplementary Table 4. We have added this figure as Supplementary Figure 6.

Revision: Supplementary Figure 6. Calculated results of different degrees of plagioclase accumulation. The proto melt composition comes from 7 mol.% fluid-present melting at 900 °C/GPa in this study. The plagioclase composition is calculated in equilibrium with the melt, based on the partition coefficient list in Supplementary Table 4.

Reviewer #4 also had a similar comment (the first main comment). In the revised text below, we therefore did our best to address both these comments from the two reviewers:

“The Palaeoarchean Barberton trondhjemite suites are suggested to have differentiated from tonalitic magma generated at <40 km depth, where plagioclase accumulation leads to a diversity of TTG, with distinct K₂O/Na₂O, Sr/Y and La/Yb but constant Ba contents³⁴. Of all rock-forming minerals likely involved in Archaean crust formation processes, only biotite exhibits strong compatibility with Ba; however, biotite accumulation during TTG melt production, ascent, and crystallization is also unlikely until the very last stages of magmatic evolution³⁴.”

I. 253-255 - see also Mathieu et al (2022) - GSA special volume on Turning Points

Revision: Good recommendation. We have cited this reference.

“The initiation of the subduction would be a function of several factors, including mantle temperature, radiogenic heat production, density contrast between the lithospheric mantle and the convective mantle, and lithospheric thickness^{36,61}.”

I. 257 - on diachronicity, would be nice to discuss this by referring to Laurent et al. (2014) - are the timing of your shifts matches this of Laurent’s shifts?

Revision: In fact, reviewer #4, Oscar Laurent himself, also recommended this reference. We have cited it.

“Subduction likely is initiated once all these factors are satisfied, which should naturally only be regional in scale at the beginning, and eventually evolve to a global scale once other regions are suitable⁶²⁻⁶⁴.”

Reviewer #4 (Remarks to the Author):

The paper by Huang et al. presents a novel concept of using Ba concentrations of Archean silicic crustal rocks as a proxy for the geodynamic setting of magma generation. They use petrological modelling to predict Ba contents of melts produced through Archean basalt melting under different geothermal gradients, and compare the results with global geochemical data on the most common Archean granitoids (TTGs) to highlight systematic variations in Ba content in these rocks. They argue that TTGs from all cratons worldwide show a shift of increasing Ba contents

with time, which they interpret as reflecting the onset of subduction; and further discuss that this shift is diachronous from craton to craton, highlighting how plate tectonics started to operate progressively on a global scale.

The topic is high-profile and suited to publication in Nature Comm. The idea of using Ba as a proxy is smart, as it is a high-concentration, widely available element even from relatively old datasets. The petrological model is well designed and the results are realistic. Altogether, the paper portrays a very reasonable and sensible view of the global geodynamic evolution throughout the Archean, consistent with that obtained using other constraints. Although the paper went already through a round of review and the authors did satisfactorily address the comments, I would have further suggestions for moderate revision and improvement before it can be accepted:

1. The strongest assumption underlying the authors' reasoning is that Ba is not affected by any possible compositional evolution of the magma after formation in the source (e.g. fractional crystallization, assimilation etc.). In turn, the authors assume that there are Ba-rich vs. Ba-poor TTG suites. In fact, one could also argue that Ba-poor vs Ba-rich TTGs represent different end-members of single TTG suites linked by differentiation processes. I don't think this is necessarily the case, but to validate the model, the authors should challenge their hypotheses against the natural TTG record. For instance, they could use the data of our recent paper on the Paleoproterozoic Barberton TTGs (Laurent et al., 2020) which, I think, strengthen their point as the least chemically evolved tonalites-diorites have similar Ba compositional range as the most evolved granitic end-member of the suite. There might be more examples to investigate in the literature (in fact, the TTG dataset from Moyen 2011 is somewhat outdated, as a lot of work has been done to understand the chemical diversity of TTGs worldwide over the past 10 years) and I encourage the authors to elaborate on this part of the discussion.

Response: We agree. The following two figures present the trace element patterns of evolved TTG and granite. Figure R1 presents the TTG and granite from Barberton TTG (Laurent et al., 2020). Figure R2 shows three groups of TTG in the Wawa domain of North America (directly cited from Kendrick et al., 2022). Even though the Sr/Y and La/Yb of these rocks show distinct patterns during magma evolution, their Ba contents are always in the same range. Figure R2 also shows the model composition during plagioclase accumulation, which is consistent with our calculation mentioned above.

Redacted

Fig. R1 Sr/Y vs La/Yb and Sr/Y vs Ba for the Palaeoarchean Barberton TTG and granite (data from Laurent et al., 2020).

Redacted

Fig. R2 Trace element patterns of three types of TTG in the Wawa domain, North America (Kendrick et al., 2022).

Revision: Reviewer #3 also had a similar comment. In the revised text below, we therefore did our best to address both these comments from the two reviewers:

“The Palaeoarchean Barberton trondhjemite suites are suggested to have differentiated from tonalitic magma generated at <40 km depth, where plagioclase accumulation leads to a diversity of TTG, with distinct K_2O/Na_2O , Sr/Y and La/Yb but constant Ba contents³⁴. Of all rock-forming minerals likely involved in Archaean crust formation processes, only biotite exhibits strong compatibility with Ba; however, biotite accumulation during TTG melt

production, ascent, and crystallization is also unlikely until the very last stages of magmatic evolution³⁴.”

2. In relation to the previous point: the major element composition of the melts produced in the models are hardly compared with those of natural TTGs – just adding natural TTG compositions to Fig 3 would already help a lot.

Revision: Excellent suggestion. We have made this suggested change (see below).

3a. About the statistical evaluation of the global TTG dataset: in terms of Ba concentrations, the real distinctive difference between the modelled low vs high geotherms is that high-Ba TTG-like melts (>800 ppm) are restricted to the low geotherms. Anything with lower Ba is basically non-discriminant. I agree with the authors that the ‘upward steps’ in the Ba time series must indicate that such high-Ba TTGs got more common in the record with time. However, there is considerable scatter in the Ba content of TTGs at a given time (often an order of magnitude or more), so it is therefore difficult to evaluate whether the steps result from the occurrence of only very few outliers at very high Ba, or a more generalized yet modest increase of Ba contents. I would love to see a more statistically rigorous way of representing the Ba range at a given time, for instance with boxes-and-whiskers indicating the interquartile range + outliers. Perhaps using median values, instead of averages, would be also more robust as the data follow a non-normal distribution. I’m pretty convinced that this would not change significantly the results, but IMHO this would make the argumentation more compelling.

Response: Firstly, it is refreshing to have a geologist reviewer that is well versed in statistics! Nonetheless, we must defend our approach as we believe it is most appropriate particularly provided the issues the reviewer has raised. Most

basically, the statistical change-point analysis we employ is designed to calculate means, not medians. So while we agree with the reviewer that provided the distribution of the Ba data, medians would be a more meaningful representation of central tendency than means, we have this practical problem to deal with. But rest assured, everything is ok, because we have converted the Ba data into log space. Plotting the data as $\log(\text{Ba})$ in a histogram reveals a good example of a log-normal distribution (i.e., normally distributed in log space). Therefore, calculating means with the change-point analysis using $\log(\text{Ba})$ is appropriate.

Revision: Because other readers also well versed in statistics may have a similar concern, we have added as Supplementary Figure 7 the log-scale histogram (shown below) that justifies—by illustrating a log-normal distribution—our conversion to $\log(\text{Ba})$ for the calculation of means using the change-point algorithm.

Supplementary Fig. 7. Histogram of global Ba data in Archaean TTG. Note log-scale on the x-axis. Red line is the kernel density estimation (KDE). The Ba data are an example of a log-normal distribution (i.e., normally distributed in log space) and are therefore converted to $\log(\text{Ba})$ for statistical analysis through time. Two outliers with low Ba values $\ll 10$ were excluded from this plot.

3b. In complement to this, from Supp Figure 5 it is like the average Ba content of some time slices is based on very few points (perhaps only one sometimes?) while others are much better constrained. Does the time-resolved Bayesian analysis include some sort of weighing to account for this? If yes, this should be explained in the methods section, and if not, it should be mentioned too. In the latter case, it might be worth running also the algorithm without the under-represented time slices and check whether the results are significantly different, or not.

Response: We greatly appreciate both the reviewer's attention to detail and their statistical mindfulness. Firstly, just to make clear that the reviewer

understands that, as stated in the figure caption, black points are individual data and blue points are averages for rocks of identical age. Therefore, considering blue “averages” that are only represented by a single datum (one black point), there are only a handful of instances like this.

Revision: We decided to retain most such individual datums, but we heed the reviewer’s larger point and chose to exclude such individual datums that were outliers, either in age (e.g., one datum in Kaapvaal that was much older than other data and based on a Nd model age) or in Ba value. Although excluding such single datums did not change our analysis much, we appreciate the reviewer making this suggestion. Because other readers also well-versed in statistics may have a similar concern, we have added the following sentence to the caption of Supplementary Figure 8:

“We excluded single datums (blue “averages” with only one black data point) that are outliers either in age and/or Ba value.”

4. Last but not least, the authors consider Ba contents as “a new compositional proxy for subduction” (l. 205). In fact, based on their model, Ba contents should be strictly regarded as a proxy for “melt production under a low geothermal gradient” – whether this resulted from subduction as we know it from modern plate tectonics, or some form of more primitive, local and/or transient process of crust burial at mantle depths (see models & papers by Taras Gerya et al., Jean Bédard et al., and discussion in Moyen & Laurent 2018) is actually a different, and unaddressed question. I guess discussing this in detail is beyond the scope of this paper but the nuance should be clearly expressed and perhaps the claim of a “proxy for subduction” somewhat tuned down.

Response: In fact, Sizova et al. (2018) have modeled the geodynamics of the Archaean ‘drip’ process, and the $P - T$ paths are still in the higher geothermal gradient field. The $450^\circ \text{C GPa}^{-1}$ would be the Archaean subduction geothermal gradient, which has been presented in the response of the 1st revision. The reviewer elaborates further in their detailed comments on Lines 199-203, to which we respond later below.

Besides these main points, I listed a few additional, minor comments in the attached commented version of the paper. I am looking forward to see this work published.

Line 24: I am not sure I fully understand what this means. I would try to rephrase **Revision:** We have rephrased this sentence for clarity:

“We conduct petrological modelling over a range of pressure–temperature conditions relevant to the Archaean geothermal gradient using an average enriched Archaean basaltic source composition to predict Ba concentrations in TTG suites, which is difficult to enhance after magma generated in the source.”

Line 98: True, but this statement somewhat hides the fact that there is a broad positive correlation between Ba and K/Na. It is in addition unclear from Fig 1 whether the high Ba TTGs really define "suites", or are just the high Ba (and possibly high K/Na ?) end of suites actually rooted at low Ba and low K/Na (see my main comment - this is something you have to discuss more in-depth, as it is a key assumption to your model).

Response: We agree that we need to clarify this, and thanks for providing evidence. As in our response to the first main comment, the Ba contents of evolved TTG and granite suites in Barberton are in the same range, although K_2O/Na_2O show obvious change (see figure below). Thus, high Ba TTG should be defined 'suites', rather than endmembers of suites rooted at low Ba and low K_2O/Na_2O .

Revision: Reviewer #3 also had a similar comment. In the revised text below, we therefore did our best to address both these comments from the two reviewers:

“The Palaeoarchaean Barberton trondhjemite suites are suggested to have differentiated from tonalitic magma generated at <40 km depth, where plagioclase accumulation leads to a diversity of TTG, with distinct K_2O/Na_2O , Sr/Y and La/Yb but constant Ba contents³⁴.”

Line 105: I largely agree, although biotite may become an important "cumulate" phase in the most evolved magmatic products (granites), as shown in Laurent et al. 2020 (see e.g. Extended Data Figures 2d and 6). I would therefore add "... is also unlikely until the very last stages of magma evolution" and refer to the above mentioned article.

Revision: We have added this sentence and refer to the paper: “Of all rock-forming minerals likely involved in Archaean crust formation processes, only biotite exhibits strong compatibility with Ba; however, biotite accumulation during TTG melt production, ascent, and crystallization is also unlikely until the very last stages of magmatic evolution³⁴.”

Line 107: Again, unless you can show that such "suites" exist I would change the wording to avoid any terminology confusion - just mentioning that there is no clear correlation between Ba and Ni, would be enough to argue against significant biotite accumulation.

Revision: Thanks for providing evidence in the comment above. We have added the citation here. See revision above.

Line 114: I would not say that TTGs are "highly enriched" – compared to modern convergent margin granitoids. TTGs actually tend to be poorer in LILE than modern arc magmas. I would rephrase to " ...too rich in LILE .to be generated from melting of MORBs "..".

Revision: We have rephrased this sentence as suggested:

"It has been noted that TTG suites are too enriched in LILEs to be generated from mid-ocean ridge basalts (MORB) and oceanic plateau basalts (OPB), which are depleted or not enriched enough in LILEs^{26,38}."

Line 124-125: If I'm not mistaking, this hypothesis has not been clearly formulated before. I guess you should first discuss the Ba compositional diversity of TTG magmas, and overall discuss whether this diversity relates to (1) distinct suites with distinct Ba contents, or (2) individual TTG suites spanning the whole Ba range (see main comment and above). Then, you may introduce one or several possible hypotheses to explain this. As written, you implicitly consider that the first possibility is correct, but I see no clear evidence from the data.

Response: Thanks for this suggestion. This comment is similar with the first main comment. We have made the change. After revision, this hypothesis should be much clearer.

Line 154: Unclear wording, do you mean "shrink the stability field of Hb and Qz to lower T"?

Revision: Yes. We have rephrased this sentence:

"Enhanced water contents also shrink the stability field of hornblende and quartz to lower temperatures, and these minerals are preferentially and efficiently consumed at relatively low temperatures (Fig. 2b)."

Line 156: Same

Revision: We have rephrased this sentence:

"For an Archaean drip setting, plagioclase stability would be shrank to lower pressure and quartz to lower temperature (Fig. 2c)."

Line 176: That is not very clearly shown in the figures. It would help to add natural TTG compositions in Fig 3 to highlight this.

Revision: We have added natural TTG compositions in Figure 3 (shown below).

Line 177-178: Why is that, actually? I mean, what in the melting phase relations of the two gradients would lead to different melt Ba contents? From what you show in Figure 2, the main difference is the relative proportion of Hbl and Grt, which makes sense, but as Ba is highly incompatible in both minerals I don't really see how it could significantly influence melt Ba concentrations. A couple of discussion sentences about this point would be welcome.

Response: The difference of Ba contents in the melt result from plagioclase in residue. One could recognize that low-degree partial melting of the source composition at a lower thermal gradient would generate Ba-rich magma, of which plagioclase would not be stable in residue, leading to a lower bulk rock $K_d(\text{Ba})$. In contrast, plagioclase would be stable in residue during low-degree partial melting at a higher thermal gradient, resulting a higher bulk rock $K_d(\text{Ba})$.

Line 179-181: OK, but as discussed by yourself above, and in the recent literature, Sr/Y ratios may not be the ideal proxy to discuss melting conditions anyway. Rather than this somewhat distractive sentence, I would prefer to see more discussion on the parameters controlling the Ba concentrations of the melts, see previous comment.

Response: See response above.

Line 182: Again, I would actually love to see the natural TTG data on Fig 3 – perhaps as shaded dots in the background, overlain by the model lines?

Revision: Revised. See new Figure 3 shown above.

Line 189: I agree this has to be a “transition”, which by definition is a progressive thing, so happens over a certain time range rather than a fixed “age”. I would rephrase to “the time period corresponding to this transition” or something similar. **Revision:** Thanks. We have edited as suggested:

“Even so, the time period corresponding to this transition is debated, with propositions ranging from the Eoarchean to Neoproterozoic^{26,48–50}.”

Line 196: I'm not sure they do - the dome-and-keel structures you are referring to in the Kaapvaal and Pilbara are assumed to be Paleoproterozoic in age (not younger than 3.2 Ga I'd say), whereas the "linear" structures in other cratons are significantly younger (mainly Neoproterozoic, say 2.5-2.7 Ga). This does not invalidate your point, but I think a better case would be that individual cratons do show a succession of "dominantly vertical" followed by "dominantly horizontal" regimes throughout their evolution (this is well documented for the Pilbara and Kaapvaal, typically).

Revision: Thanks. We have made the edits as suggested:

“Well-preserved vertical structures can be recognized at some older cratons, such as eastern Pilbara cratons in western Australia, Kaapvaal and Zimbabwe cratons in southern Africa^{51–53}, although individual cratons do show a succession of ‘dominantly vertical’ followed by ‘dominantly horizontal’ regimes throughout their evolution.”

Line 199-203: From your modelling, I agree with these conclusions. However, the relevant question is then : how common are Ba-rich TTGs in the Archean record? In particular, are the Ba-rich melts specific to your "cool" gradients that common? (see main comment about this)

Response: We are happy to hear that the reviewer agrees with these conclusions on the basis of our modeling. To their question concerning the prevalence of Ba-rich melts specific to “cool” gradients in the Archean, this is specifically addressed in the following section (starting at Line 224 in the previous submission) by statistically analyzing the Ba contents of Archean TTG.

Line 205: This sounds a bit ambitious (see my main comment about this point), perhaps reformulate to tune this down.

Response: See response to the last main comment.

Line 221: Again, for low-geothermal gradient melting, yes – but for “subduction”, I am not sure. It all depends what we mean by “subduction”!

Response: See response to the last main comment.

Line 231-232: I agree, but even if they are not outliers, they may represent only a subordinate proportion of the dataset (on figure 4, maybe 80% or more of the points shown have $\log(\text{Ba}) < 3$, which would not make them high-Ba TTGs in the sense of your definition. Moreover, and this joins previous comments, you would need to carefully assess whether this high-Ba end-member represents one component of otherwise Ba-poorer TTG suites; or truly is a characteristic of entire suites. **Response:** This comment is a mix of previous ones addressed specifically above.

Line 236-237: I only partly agree - yes, there is a big step at 3.7 Ga, but I'd say there is a smaller one at 3.2 Ga or so. In fact, it is from 3.2 Ga on that truly "high-Ba" (>1000 ppm) TTGs seem to become significantly abundant in the record, which makes your point. Again, perhaps a more rigorous statistical evaluation of the global TTG dataset would help to highlight this? (see main comment)

Response: Again, we appreciate the reviewer's attention to the details of the data and the statistical analysis. Indeed, the reviewer is correct that if the change-point algorithm is allowed to have multiple change points instead of just one, another two increases in Ba occur at ca. 3.1 Ga and 2.8 Ga (shown below).

Revision: Because the reviewer suggested we added new data for Kaapvaal (next comment below), we also had to reanalyze the global dataset as well as that of Kaapvaal. Interestingly, the multi-step interpretation completely supports the reviewer's expectation of an additional younger shift as well as our interpretation that the onset of subduction is globally diachronous. The modified Figure 4 is shown below.

Line 240: Very importantly, I guess what is reported as "Kaapvaal craton" in there only concerns the eastern Kaapvaal (Barberton region), as there are very few data points younger than 3.2 Ga. I would add the 3.3-2.8 Ga TTG data from the northern Kaapvaal (see the compilation in Laurent et al. 2019b, https://doi.org/10.1007/978-3-319-78652-0_4), which would probably show your positive step at about 3 Ga and thus neatly fit your global curve in Fig 4b.

Revision: We are very thankful to the reviewer for alerting us to the availability of more data of younger TTG from Kaapvaal craton. We have downloaded the reviewer's data from Laurent et al. (2019b) and redone our analysis of both Kaapvaal craton and globally. Whereas before Kaapvaal showed an unexpected decrease in Ba (which would contract our model), the new dataset with younger TTG data included exhibits a flat pattern (shown below, the modified Kaapvaal panel of Supplementary Figure 8) similar to the Slave craton, with both Slave and

Kaapvaal starting their TTG rock records with Ba elevated above $\log(\text{Ba}) = 2.5$, which for a craton with such old rocks, is already elevated. As anticipated by the reviewer, such a modified result for Kaapvaal is consistent with the oldest-rock– Ba-shift correlation by craton in Figure 4b to which Kaapvaal has now been added. Both panels of Figure 4 (shown above) and the Kaapvaal panel of Supplementary Figure 8 (shown below) have been updated thanks to the informed reviewer’s excellent advice. (Note that considering a single datum at 3.7 Ga for Kaapvaal would yield a better fit in Figure 4b, but this datum is an outlier in age, a Nd model age, and older than the oldest rocks of Kaapvaal according to Moyen et al. (2007), so we deem it unreliable and exclude it from our analysis.)

Moyen, J.-F., Stevens, G., Kisters, A.F.M., Belcher, R.W., 2007. TTG plutons of the Barberton granitoid-greenstone terrain, South Africa. In: van Kranendonk, M.J., Smithies, R.H., Bennett, V. (Eds.), Earth’s Oldest Rocks. Elsevier, pp. 606-668.

Line 247: shifts

Revision: We have changed it.

Line 256-260: Yes, I fully agree with you in this respect. In fact, we have already proposed this model based on other constraints (i.e. the general evolution of granitoid rock types with time in different cratons) in Laurent et al. (2014) (<https://doi.org/10.1016/j.lithos.2014.06.012>) and Laurent et al. (2019a) (<https://doi.org/10.1016/j.precamres.2018.12.020>). These papers should be referred to here.

Revision: Thanks for recommendation. We have added the citations:

“Subduction likely is initiated once all these factors are satisfied, which should naturally only be regional in scale at the beginning, and eventually evolve to a global scale once other regions are suitable⁶²⁻⁶⁴.”

Fig.3: Would be good to label some of them, at least the first and last one; or add arrows to indicate the direction of compositional melt evolution with increasing melt fraction.

Response: In fact, we have already made such a distinction in the direction of compositional melt evolution. The background color gradient indicates the temperature increase (yellow to orange).

Fig.4: Two problems with this figure :

- The "step changes by craton" panel is confusing, because it partly overlaps with the y-axis of the $\log(\text{Ba})$ vs time plot, so one may misunderstand that the depicted evolutions are at very high Ba content;
- As far as I can tell the axes of panel b are swapped.

Revision: First, thanks for pointing this potential confusion out; we have moved the "step changes by craton" higher so that it does not overlap with the y-axis of the $\log(\text{Ba})$ vs. time plot. Second, good catch! We have swapped the axes labels.

Sincerely,

Oscar LAURENT - Toulouse, 07.07.2022

REVIEWERS' COMMENTS

Reviewer #4 (Remarks to the Author):

I enjoyed reading the revised version of the manuscript by Huang et al., dealing with the use of Ba contents of Archean TTGs as a proxy for melting conditions. The authors have done a great job in improving the manuscript according to reviewers' comments, which definitely strengthen their case. Further questions came to my mind as the revisions highlighted intriguing aspects of the data (e.g. mismatching model results and TTG data in terms of Mg and Ni contents in Fig 3); but these are second-order features that do not challenge the paper's main message about Ba, and discussing them would definitely be side-tracking. Therefore, the manuscript can now be accepted, after only very minor formal revision as suggested below.

- L. 23-24: Even after rephrasing I do not understand well what you mean by "which is difficult to enhance after magma generated in the source". Do you mean that unlike other trace elements, Ba contents are relatively insensitive to process of magma evolution upon transfer and emplacement?
- L. 81, 95, 101, 158: "enhance" is misused I think – I guess this is the reason for the previous comment too. I am not a native English speaker so maybe I'm wrong, but to me "enhance" has a more qualitative meaning, synonym to "improve" or "upgrade", and thus is not best suited to describe a measurable change of a physical variable. I'd rather use "increase" or "raise".
- L. 102-103: I would begin this sentence by "For example" or "As an illustration", to smoothen the transition to the Barberton case study.
- L. 142: I'd change "most accepted" to "two possible end-member" – actually none of these settings is well accepted, and may truly reflect the situation in the Archean!
- L. 223-226: These two sentences sound a bit awkward and redundant. I would delete the first one.
- L. 238 and 245-246: From what I understand of your revisions, the Kaapvaal craton TTGs also show homogeneously high Ba as the Slave ones; better be mentioned for clarity.
- L. 239-240: Perhaps replace "in an Archaean drip setting" by "along a hot geotherm", which is less oriented in terms of geodynamic setting (see l. 142)
- L. 263: "other regions are suitable": a little unclear, perhaps better "all regions eventually became favorable to subduction initiation".

I apologize again for the delay in reviewing and am looking forward to see this work published.
Sincerely, Oscar LAURENT – Toulouse, 07.10.2022

Responses in blue

Revisions in red

REVIEWERS' COMMENTS

Reviewer #4 (Remarks to the Author):

I enjoyed reading the revised version of the manuscript by Huang et al., dealing with the use of Ba contents of Archean TTGs as a proxy for melting conditions. The authors have done a great job in improving the manuscript according to reviewers' comments, which definitely strengthen their case. Further questions came to my mind as the revisions highlighted intriguing aspects of the data (e.g. mismatching model results and TTG data in terms of Mg and Ni contents in Fig 3); but these are second-order features that do not challenge the paper's main message about Ba, and discussing them would definitely be side-tracking. Therefore, the manuscript can now be accepted, after only very minor formal revision as suggested below.

- L. 23-24: Even after rephrasing I do not understand well what you mean by "which is difficult to enhance after magma generated in the source". Do you mean that unlike other trace elements, Ba contents are relatively insensitive to process of magma evolution upon transfer and emplacement?

Revision: We have changed "enhance" to "increase" as the following comment.

- L. 81, 95, 101, 158: "enhance" is misused I think – I guess this is the reason for the previous comment too. I am not a native English speaker so maybe I'm wrong, but to me "enhance" has a more qualitative meaning, synonym to "improve" or "upgrade", and thus is not best suited to describe a measurable change of a physical variable. I'd rather use "increase" or "raise".

Revision: We have changed them to "increase" or "raise".

- L. 102-103: I would begin this sentence by "For example" or "As an illustration", to smoothen the transition to the Barberton case study. **Revision:** We have changed it.

- L. 142: I'd change "most accepted" to "two possible end-member" – actually none of these settings is well accepted, and may truly reflect the situation in the Archean!

Revision: We have changed it.

- L. 223-226: These two sentences sound a bit awkward and redundant. I would delete the first one.

Revision: We have deleted the first one.

- L. 238 and 245-246: From what I understand of your revisions, the Kaapvaal craton TTGs also show homogeneously high Ba as the Slave ones; better be mentioned for clarity.

Revision: We have mentioned it.

- L. 239-240: Perhaps replace “in an Archaean drip setting” by “along a hot geotherm”, which is less oriented in terms of geodynamic setting (see l. 142)

Revision: We have changed it.

- L. 263: “other regions are suitable”: a little unclear, perhaps better “all regions eventually became favorable to subduction initiation”.

Revision: We have changed it.

I apologize again for the delay in reviewing and am looking forward to see this work published. Sincerely, Oscar LAURENT – Toulouse, 07.10.2022